# Epstein-Barr virus LMP1 manipulates the content and functions of extracellular vesicles to enhance metastatic potential of recipient cells

**Dingani Nkosi**[ID][ɵ], **Li Sun**[ID][ɵ], **Leanne C. Duke**[ID], **David G. Meckes, Jr.** *

Department of Biomedical Sciences, Florida State University College of Medicine, Tallahassee, Florida, United States of America

ɵ These authors contributed equally to this work.

* david.meckes@med.fsu.edu

**Data Availability Statement:** All relevant data are within the manuscript and its Supporting Information files. The genomics data discussed in this publication have been deposited in NCBI's

## Abstract

Extracellular vesicles (EV) mediate intercellular communication events and alterations in normal vesicle content contribute to function and disease initiation or progression. The ability to package a variety of cargo and transmit molecular information between cells renders EVs important mediators of cell-to-cell crosstalk. Latent membrane protein 1 (LMP1) is a chief viral oncoprotein expressed in most Epstein-Barr virus (EBV)-associated cancers and is released from cells at high levels in EVs. LMP1 containing EVs have been demonstrated to promote cell growth, migration, differentiation, and regulate immune cell function. Despite these significant changes in recipient cells induced by LMP1 modified EVs, the mechanism how this viral oncogene modulates the recipient cells towards these phenotypes is not well understood. We hypothesize that LMP1 alters EV content and following uptake of the LMP1-modified EVs by the recipient cells results in the activation of cell signaling pathways and increased gene expression which modulates the biological properties of recipient cell towards a new phenotype. Our results show that LMP1 expression alters the EV protein and microRNA content packaged into EVs. The LMP1-modified EVs also enhance recipient cell adhesion, proliferation, migration, invasion concomitant with the activation of ERK, AKT, and NF-κB signaling pathways. The LMP1 containing EVs induced transcriptome reprogramming in the recipient cells by altering gene expression of different targets including cadherins, matrix metalloproteinases 9 (MMP9), MMP2 and integrin-α5 which contribute to extracellular matrix (ECM) remodeling. Altogether, our data demonstrate the mechanism in which LMP1-modified EVs reshape the tumor microenvironment by increasing gene expression of ECM interaction proteins.

## Author summary

Extracellular vesicles (EV) facilitate cell-to-cell crosstalk due to their capability to sort and transfer various cargoes. Multiple studies have shown changes in EV content and cargo

Gene Expression Omnibus (103) and are accessible through GEO Series accession number GSE155202. (https://www.ncbi.nlm.nih.gov/geo/query/acc.cgi?acc=GSE155202) Proteomics data files have been submitted to ProteomeXchange with identifier PXD021914 (http://proteomecentral.proteomexchange.org/cgi/GetDataset?ID=PXD021914).

**Funding:** This study was supported by grants from the National Cancer Institute of the National Institutes of Health (NCI.gov and NIH.gov) grant numbers RO1CA204621 and R15CA188941 awarded to DM. The funder did not play any role in the study design, data collection and analysis, decision to publish, or preparation of the manuscript. The content is solely the responsibility of the authors and does not necessarily represent the official views of the National Institutes of Health.

**Competing interests:** The authors have declared that no competing interest exist.

affect their functions and contribute to pathological conditions such as cancer. EVs represent a mechanism through which cancer cells modify their microenvironment to enhance growth and metastasis. This study showed that LMP1, an EBV major oncoprotein which is released in EVs alters the EV content and cargo leading enhanced cell attachment, proliferation, migration and invasion. LMP1 modified EVs mediate the transfer of signaling molecules to recipient cells where they induce NF-κB, AKT and MAPK/ERK signaling pathways leading to alteration in gene expression especially those involved in ECM interaction. LMP1 containing EVs modify the microenvironment by upregulating cadherins, fibronectin, integrin-α5, MMP9 and MMP2 to promote a tumor permissive niche leading tumorigenesis or metastasis.

## Introduction

Epstein-Barr virus (EBV) persistently infects over 90 percent of the world's population, with an estimated 200,000 new cancers each year which is approximately 2% of all cancers world-wide [1,2]. EBV latent infection is associated with development of cancers such as nasopharyngeal carcinoma (NPC), Burkitt lymphoma, Hodgkin's disease, and posttransplant lymphomas especially in immunocompromised or genetically susceptible individuals [3,4]. Latent membrane protein 1 (LMP1) is the chief EBV oncogene which is expressed in most EBV associated cancers [5–7].

LMP1 is a CD40 receptor mimicry that contains N-terminal cytoplasmic domain, six transmembrane and cytoplasmic C-terminal domain which harbors the C-terminal activating regions (CTAR1, 2, and 3) [6,8–12]. LMP1 activates many signal cascades including mitogen-activated protein kinase/extracellular signal-regulated kinase (MAPK/ERK), phosphatidylinositol 3-kinase (PI3K)/AKT, NF-κB, STAT3, mTOR, epidermal growth factor receptor (EGFR) and c-Jun N-terminal kinase (JNK) through the interaction of tumor necrosis factor receptor-associated factors (TRAFs) and other effector molecules to CTARs [13–15]. These LMP1-activated pathways are known to induce the expression of many downstream products that influence cell growth, apoptosis, migration, and invasion [9,14,16–18]. In case of EBV associated cancers, activation of these pathways by LMP1 results in tumorigenesis and metastasis [5,19]. LMP1 plays an important role in tumorigenesis and metastasis by inducing epithelial-mesenchymal transitions (EMT) and its associated cell adhesion, motility and invasion features [20–26]. Induction of the EMT is mediated via CTAR1 domain through integrin-mediated ERK-MAPK signaling which can cause a downstream cadherin switch [20,27,28]. LMP1 downregulates E-cadherin and upregulates N-cadherin through mechanism that involve transcriptional repression of Twist and Snail [23]. Furthermore, expression of LMP1 has been shown to upregulate type IV collagenase matrix metalloproteinase-9 (MMP9) and MMP1 which are responsible for destruction of the ECM [29–31]. Additionally, cellular expression of LMP1 also induces Hypoxia-inducible factor-1α (HIF1α), a transcription factor which is associated with enhanced invasion and angiogenesis [32–34]. Taken together, these data support the notion that LMP1 plays an important role in remodeling of the tumor microenvironment to promote metastasis. However, mechanisms of how LMP1 manipulates the tumor microenvironment remodeling are still not completely understood.

Intercellular communication plays a vital role in both normal and pathological processes. In case of cancer, communication between tumor cells and their surrounding microenvironment is important in development and progression of cancer. Apart from soluble factors like cytokines, extracellular vesicles (EVs) have more recently been shown to modulate cell-to-cell communication in normal physiological processes as well as in pathological conditions, such

as cancer [35,36]. EVs are a heterogenous population of membrane enclosed vesicles comprised of exosomes, microvesicles and apoptotic bodies grouped according to size, sub-cellular origin, molecular content and density [37–40]. EVs package and transfer biologically active cargo including proteins, mRNAs, microRNAs (miRNAs), and lipids to neighboring or distant cells [41–45]. Numerous studies have reported the significant role of EVs play in cell growth, invasion, and metastasis of diverse cancers [46,47]. EBV infected cells release EVs that can contain the viral proteins, LMP1 and LMP2, and virally encoded miRNAs [42,48–50]. EBV has been demonstrated to modify protein content and cargo of EVs released from latently infected B-cells with most of the significant changes correlating to LMP1 expression [51]. Cellular expression of LMP1 enhances vesicle release and the LMP1-modified EVs can activate downstream signaling cascades including MAPK/ERK and PI3K/AKT in recipient cells [49,52–54]. LMP1 containing EVs promote tumor cell proliferation, migration, invasion potential, and promote radio resistance of NPC [32,55–57]. Various cargo has been found to be packaged into the LMP1 containing vesicles including EGFR, fibroblast growth factor (FGF-2) and HIF1α which play major roles in angiogenesis, tumor growth and metastasis [32,48,58]. These studies demonstrate that LMP1 modified EVs support an establishment of a tumor permissive microenvironment hence promoting cancer development and metastasis.

Though the mechanism is not well understood, we speculate that LMP1 containing EVs are taken up by the recipient cells and activate cell signaling pathways and target gene expression which modulates the biological properties of recipient cell towards a new phenotype. Recently it has been shown that EVs from Kaposi's Sarcoma-associated herpesvirus (KSHV) infected cells can rewire gene expression of the recipient non-infected cells to promote cell proliferation and migration [59]. In this study, we demonstrate that cellular expression of LMP1 alters the EVs cargo and content which leads to promoting cell attachment, proliferation, migration and activation of ERK and AKT pathways of the recipient cells treated with the vesicles. Exposure of the recipient cells to the LMP1 modified EVs resulted in changes in gene expression especially ECM associated proteins. LMP1 modified EVs upregulated the gene expression of cadherins, MMP9, MMP2, fibronectin and integrin-α5 in the recipient cells. Further analysis showed that LMP1 EVs promote cell attachment through integrin-α5. Additionally, these LMP1 containing EVs exhibit increase in MMP activity and cell invasion assays. Altogether, these results begin to unfold the mechanisms LMP1 modified EVs utilizes to remodel the tumor microenvironment through increased gene expression of the ECM interacting proteins.

## Results

### Enrichment and characteristics of LMP1 modified EVs

Different methods have been employed to separate and purify EVs from cell culture supernatant or plasma. Some of the commonly utilized methods include: ultracentrifugation (UC), tangential flow filtration (TFF), polyethylene glycol (PEG) precipitation, sucrose or iodixanol density gradient ultracentrifugation, antibody-based bead capture and size exclusion chromatography [60–63]. For large amounts of media, a combination of these methods can be used for purification of the EVs. An ideal purification method would retain EV functional properties [60,61]. In the case of LMP1 modified EVs, we have previously shown that expression of this viral oncoprotein enhances the release of smaller EVs in different cell lines with an enrichment of different exosomal markers [49,50]. In this study, the EVs used were isolated by a combination of tangential flow filtration (TFF, 100kDa cutoff) followed by precipitation using PEG-6000 then ultracentrifugation to remove PEG and contaminating protein complexes (Fig 1A). Previous studies have shown that isolation of EVs using a combination of these methods keeps EV loss at a minimum and the purified EVs still maintain the biophysical properties

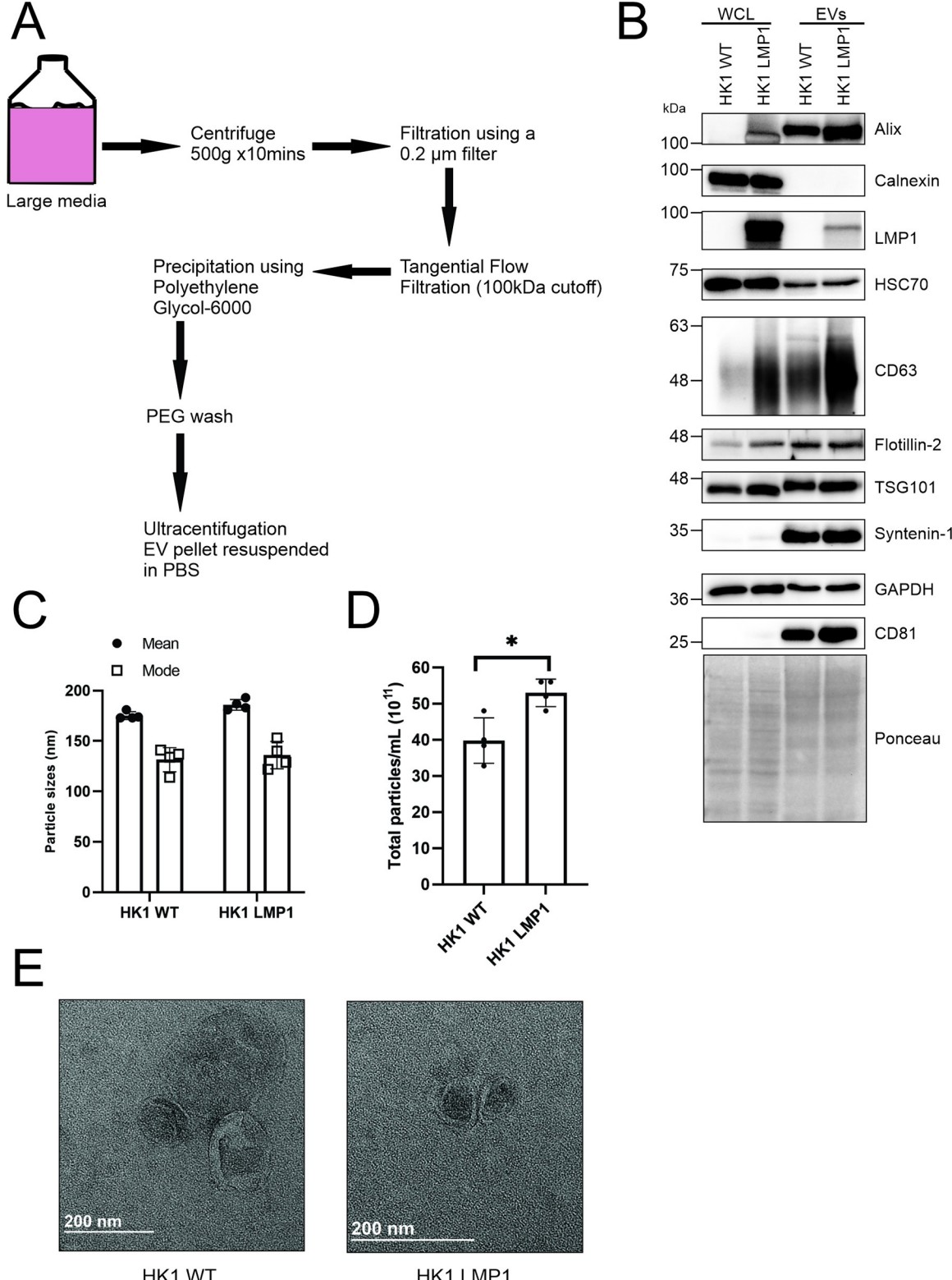

**Fig 1. Isolation and characterization of EVs.** EVs were harvested from HK1 WT and HK1 expressing LMP1 cells using a combination of UC, TFF and PEG. (A) Schematic diagram of EV isolation from conditioned media of HK1 WT or HK1 LMP1 cells. (B) Western blot analysis of cell and EV lysates from wild type HK1 and HK1 expressing inducible LMP1 showing expression levels of the different EV markers including CD63, flotillin-2, TSG101, syntenin-2, CD81 and calnexin. WCL: whole cell lysate (C-D) The isolated EVs was

analyzed by Nanoparticle Tracking Analysis for the sizes and quantity. (E) Negative staining electron microscopy of the HK1 WT and HK1 LMP1 EVs. *, P < 0.05.

[59–61]. To begin to characterize the EVs using MISEV2018 guidelines [64], the EVs were harvested from either HongKong1 (HK1) wild type cells (a nasopharyngeal carcinoma cell line) or HK1 cells containing an inducible LMP1 construct. Immunoblot blot analyses of the cell and vesicle lysates looking at different EV related protein markers including CD63, CD81, Alix, TSG101, Syntenin-1, HSC70 and Calnexin were performed (Fig 1B). The immunoblots showed enrichment of different protein markers in the LMP1 containing vesicles. The vesicles isolated from both samples had a mean particle size of < 200nm and a mode of < 150nm, befitting to be called small EVs (Fig 1C). As expected, the nanoparticle tracking analysis revealed an increase in release of particles/mL when LMP1 was expressed in the cells compared to the wild type (Fig 1D). Lastly, we validated the EVs by negative staining electron microscopy which showed cup shaped vesicles less than 200nm (Fig 1E). Taken together our results revealed that the method the used isolated and enriched for EVs.

## LMP1 expression modifies EV cargo and content

LMP1 has been suggested to be one of major players in the modification of the EV proteome in context of an EBV infection [48,51]. Mass spectrometry data analysis on EVs purified from patient derived B cell lines either uninfected or infected with EBV, KSHV or dually infected showed LMP1 containing EVs displayed a unique clustering pattern according expression levels of LMP1 [51]. To further assess whether the modification of the EV content and cargo was unique to LMP1 and no other EBV associated proteins, we performed label free proteomics on EVs collected from either HK1 WT or HK1 overexpressing LMP1 cells. For these proteomics experiments, highly purified EVs were obtained following Optiprep density gradient ultracentrifugation [60]. Using this method we identified about 1600 total proteins from three independent biological replicas with about 140 proteins unique to the vesicles containing LMP1 (S1 Data) (Fig 2A). From the dataset we identified 300 proteins to be 2-fold upregulated and 137 proteins to be downregulated by LMP1. The 300 proteins which were 2-fold upregulated by LMP1 were subjected to bioinformatic analysis. Cellular component analyses revealed that majority of the proteins associated with exosomes and lysosomes (Fig 2B). LMP1 has been shown to traffic through the endocytic pathways to lysosomes for degradation or exosomes for release [50,52]. Pathway analysis demonstrated that most proteins were involved in EBV infection, endocytosis, apoptosis, cell adhesion molecules, MAPK signaling pathway, tumor necrosis factor (TNF), NF-κB and HIF-1 signaling (Fig 2C). Lastly, analysis of the biological processes showed an enrichment in localization establishment, transportation, viral production, regulation of signal transduction, immune response, viral reproduction and cell migration (Fig 2D). This was not surprising as most of the LMP1 interacting proteins previously identified are involved in these pathways and biological processes. Bioinformatic analyses of the downregulated proteins showed an enrichment of organelle organization, protein localization, catabolic process and cell death in biologic processes (S1A and S1B Fig). Comparison to the Vesiclepedia database revealed that almost 95% of the identified proteins have previous been found in EVs (Fig 2E). The identified proteins including metalloproteinases, intergrins, and EV biogenesis markers were validated via immunoblots for their expression in EVs. (S1C Fig). LMP1 expression in the EVs increased expression of the different identified genes compared to the wild type. Enrichment of the different pathways and biological processes also demonstrated the versatility of LMP1 of being involved in multiple cellular processes and the wider protein networks of the viral oncoprotein [65].

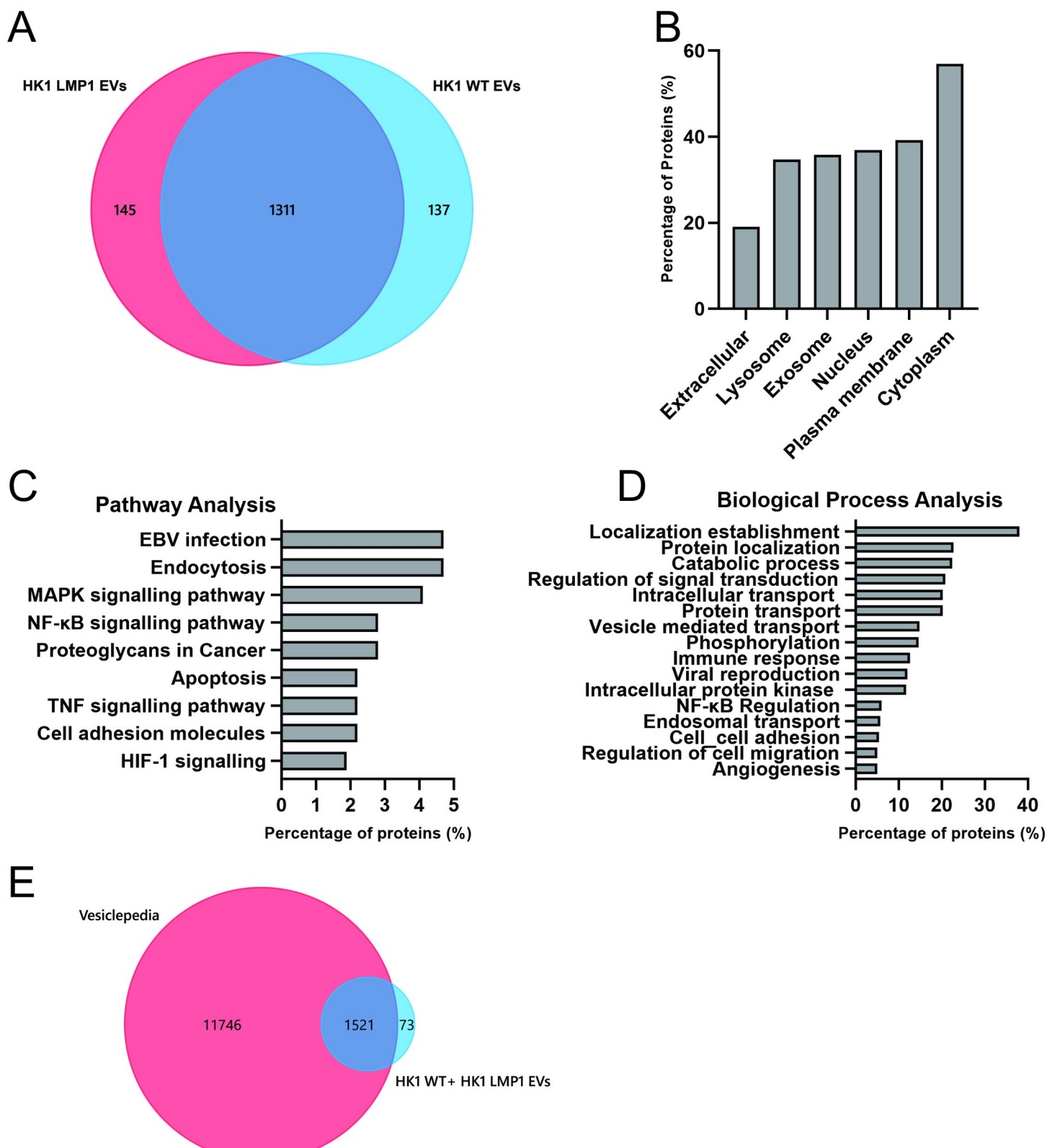

**Fig 2. LMP1 modifies EV protein cargo and content.** Mass spectrometry data analysis of the HK1 WT and HK1 LMP1 EVs. (A) Venn diagram of the identified proteins common and unique to HK1 WT and HK1 LMP1 EVs. Enrichment analysis of the identified proteins which were 2-fold upregulated by LMP1. (B) Cellular compartmentalization of proteins in the dataset was examined using FunRich. (C) Pathways (KEGG) and D) biological processes analysis (BP_DIRECT) of the LMP1 upregulated proteins identified was done using Networkanalyst 3.0. (E) Venn diagram of identified proteins compared to EV proteins found in Vesiclepedia.

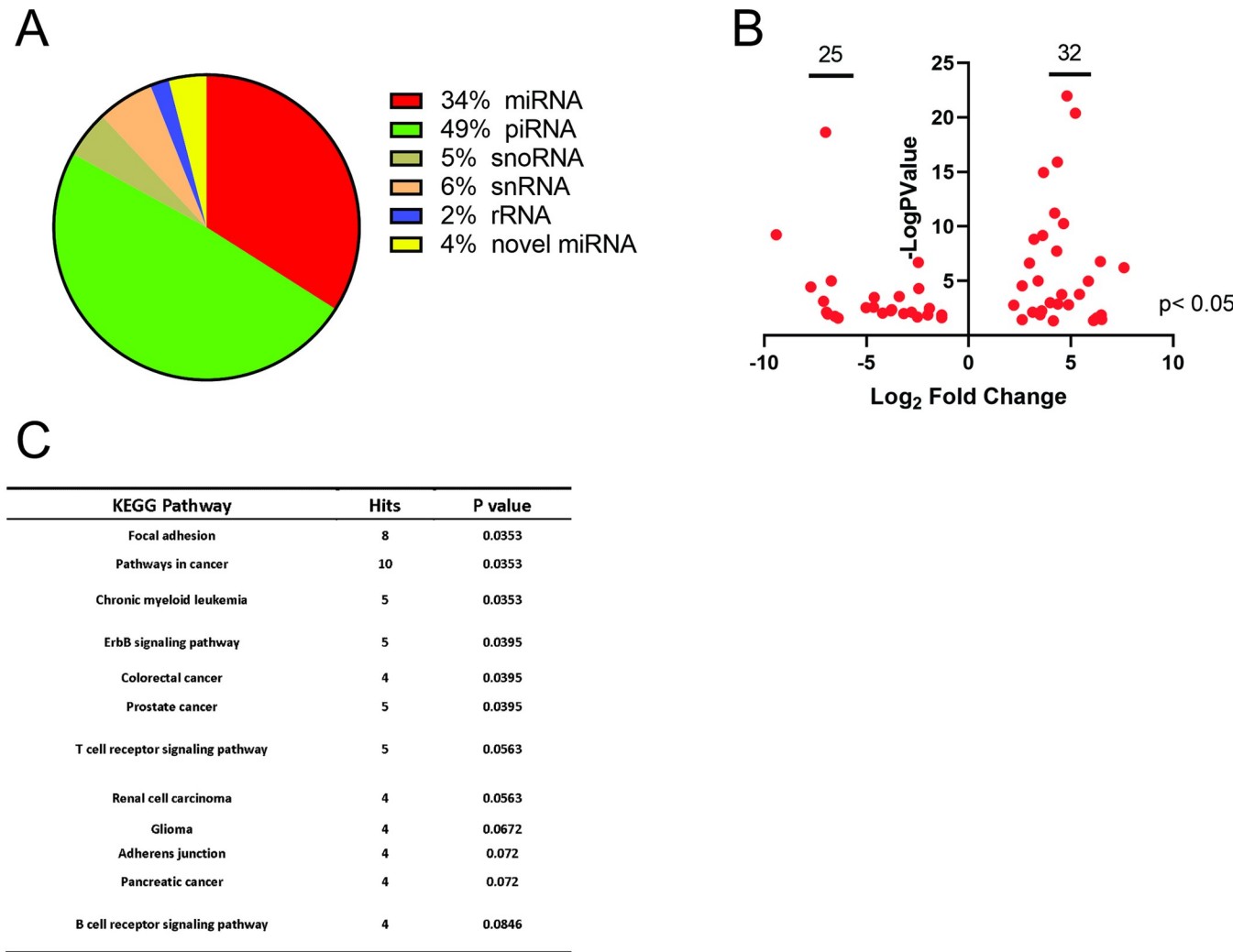

**Fig 3. LMP1 expression alters miRNA packaged in EVs.** Small RNAs were isolated from HK1 WT and HK1 LMP1 EVs. (A) Pie chart of differential expression of small RNA by category. (B) Volcano plot of the miRNA which were Log2-Fold upregulated or downregulated. (C) KEGG pathway analysis of the identified miRNAs.

EVs have been also been shown to package small RNAs which are biologically active and affect processes such as tumor immunity, growth, invasion and angiogenesis [42,43,66,67]. Since LMP1 has been shown to alter the cellular expression of different small RNAs, we assessed whether LMP1 expression will alter the small RNAs especially miRNAs packaged into the EVs [44,68]. Standardized protocol was performed with minor modifications to maximize miRNA recovery and ligation efficiency for RNA-seq as described in methods. Our results showed piwi-RNA (~50%) as the most abundant read followed by miRNA which consisted of about 34% of total reads (Fig 3A). Similar distribution of the small RNAs has been reported by different research groups. Compared with control HK1 WT EVs, LMP1 expression modified EV miRNAs cargo packaged. We identified about 32 upregulated and 25 downregulated miR-NAs (S2 Data) which were statistically significant (p<0.05) and most of them were changed dramatically with Log2-fold change more than 5 (Fig 3B). Pathway analysis of the differentially expressed miRNAs identified an enrichment in focal adhesion, pathways in cancer and adherens junction (Fig 3C). The predicted protein hits from these identified pathways included

MAPK1, PTEN, GSK3B and CRKL which have been shown to be affected by cellular expression of LMP1 (S1 Table). Taken together, LMP1 can modify the EV proteome and alter the miRNAs packaged into the EVs which might influence the function of the EVs on recipient cells.

## LMP1 modified EVs promote cell adhesion, proliferation, migration and can activate AKT and ERK pathways

LMP1 expression induces EMT and its associated cell adhesion, motility and invasion features in rodent fibroblasts and epithelial carcinoma cell lines [20,21,26]. In pre-malignant cell line MCF10A, LMP1 has also been shown to increase cell adhesion, migration and motility [20]. Currently, different studies indicate that LMP1 might be playing a major role in remodeling of the tumor microenvironment through the transfer of virally-modified EVs leading to tumor growth, immune cell regulation, and metastatic processes. LMP1-modified EVs enhance tumor proliferation, migration, invasion potential, and promote radio resistance of nasopharyngeal carcinoma [32,55,56,69]. To assess the functional capacity the LMP1 modified EVs isolated play in enhancing cell adhesion, proliferation and migration of epithelial cells, we monitored these different phenotypes using xCelligence system which uses electrical impedance. Our initial studies confirmed that expression of LMP1 in HK1 cells enhanced cell attachment compared to HK1 WT (S2A Fig). Interestingly, HK1 cells expressing LMP1 C-terminal activating region (CTAR) 1 promoted rapid cell attachment compared to CTAR 2 expressing cells (S2B Fig). Furthermore, LMP1 modified EVs were shown to increase cell attachment of the HK1 WT cells compared to HK1 WT EVs (S2A Fig). In transformed nasopharyngeal carcinoma cells like HK1, the expression of some of the genes are likely up-regulated to a saturated level, therefore exposure to LMP1 modified EVs may not result in higher levels of attachment or motility. Therefore, we evaluated the role of LMP1 modified EVs in enhancing cell attachment of MCF10A cells, a non-tumorgenic epithelial cell line, which were either incubated with HK1 WT EVs or PBS. The results revealed that the LMP1 modified EVs enhanced the cell attachment of the MCF10A cells compared to the HK1 WT EVs or PBS (Fig 4A and 4B). Comparative analysis showed that HK1 cells expressing LMP1 enhanced cell adhesion more rapidly compared to MCF10A cells treated with EVs (S2C Fig). Furthermore, our results showed that enhancement of the MCF10A cell attachment by the EVs can be dose dependent (S2D Fig). LMP1 modified EVs also increased the cell proliferation of recipient cells compared to the HK1 WT EVs (Fig 4C and 4D). Lastly, we assessed cell migration by incubating the MCF10A cells with EVs in top chamber and media with a 2–3% FBS chemoattractant in the lower chamber. LMP1-modified EVs significantly promoted cell migration of the recipient cells (Fig 4E and 4F). Altogether, our results demonstrate the pro-migratory phenotype effects LMP1 modified NPC EVs are inducing in recipient cells.

Transfer of the LMP1 containing EVs through paracrine or autocrine mechanism can activate MAPK/ERK and PI3K/AKT in the recipient cells which is important for LMP1-mediated stimulation of growth signaling pathways [48,57]. To test the potential of the LMP1 modified EVs in activating the ERK and AKT pathways, MCF10A cells were exposed to the EVs in serum free conditions. Our results showed that the LMP1 containing EVs activated both the AKT and ERK pathways higher compared to HK1 WT EVs (Fig 5A). Additionally, we evaluated whether the LMP1 modified EVs would enhance activation of the NF-κB pathway more than the HK1 WT EVs. To test this, MCF10A cells expressing the NF-κB luciferase reporter were made and exposed to the EVs in serum free conditions. LMP1 modified EVs enhanced the NF-κB activation compared to HK1 WT EVs however it was not statistically significant (Fig 5B). This could be due to the already high NF-kB activation of EVs from the tumorgenic

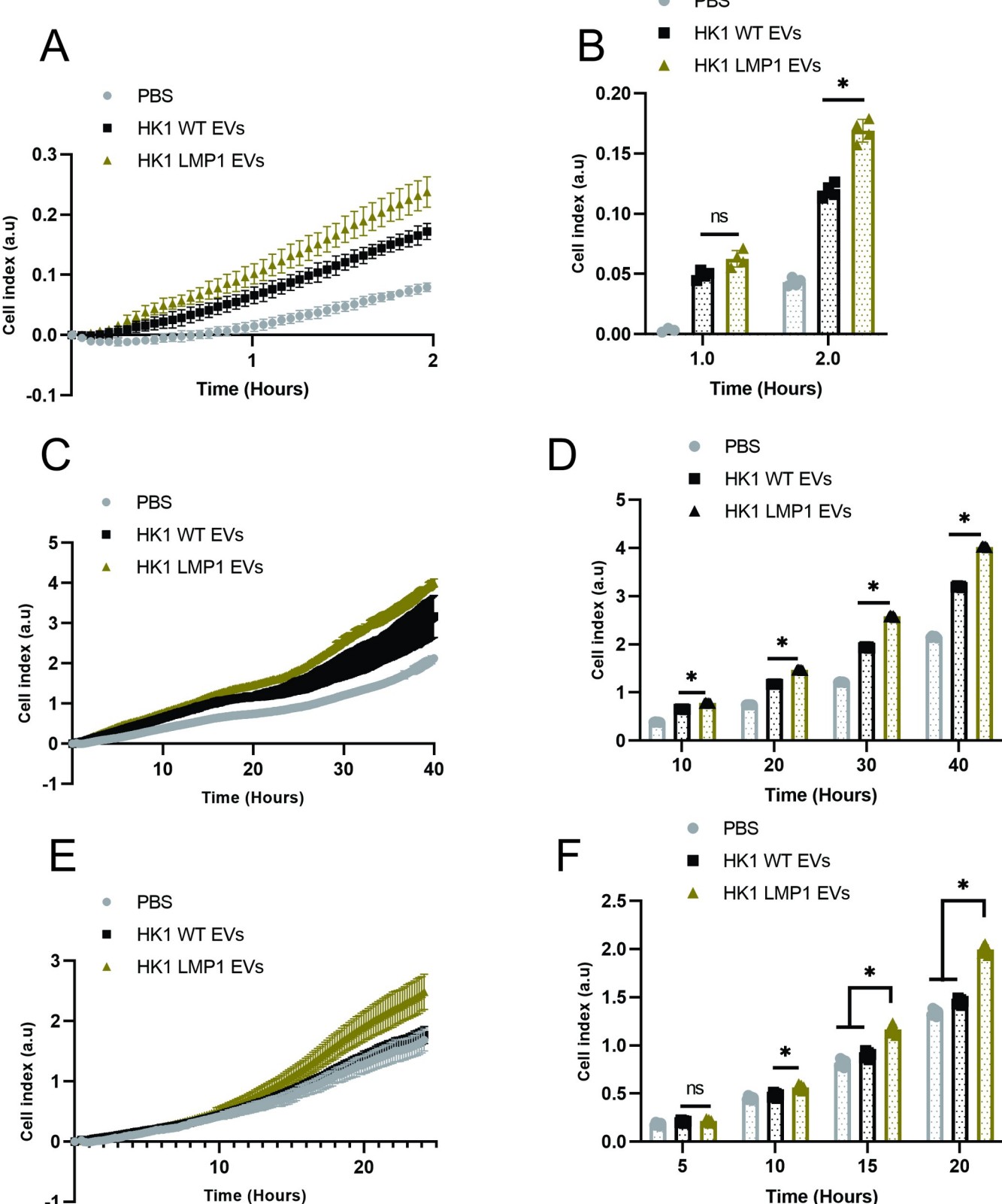

**Fig 4. LMP1 modified EVs enhance cell adhesion, proliferation and migration.** MCF10A cells were exposed to equal protein of HK1 WT, HK1 LMP1 EVs or equivalent volume of PBS prior to seeding them into xCelligence E-16 or CIM-16 plate. Cell attachment, proliferation and migration were monitored for

about 2, 40 and 24 hrs., respectively. (A-B) LMP1 modified EVs promote cell attachment of the MCF10A. (C-D) LMP1 containing EVs promote cell proliferation and growth. (E-F) LMP1 modified EVs promotes cell migration. Bar charts (B, D, F) show the indicated time points of corresponding growth curve (A, C, E). *, P < 0.05.

HK1 cells. Taken together, these results show the capacity of the LMP1 modified EVs in enhancing activation of signaling pathways which might be important in the activation of different downstream cellular processes including cell attachment, growth, migration, and invasion.

### HK1 WT and HK1 LMP1 EVs reprogram gene expression of recipient cells

To further understand the molecular mechanisms underlying the changes in phenotypes observed in the MCF10A cells by the LMP1 modified EVs, RNA-seq was performed on the recipient cells. In normal physiological condition or tumor microenvironment, cells are constantly exposed to EVs circulating which can be taken up by these cells and rewire the cells towards a different phenotype [59]. To depict this, MCF10A cells were exposed to HK1 WT EVs, HK1 LMP1 EVs or PBS for 48 hours, new EVs were added every 24 hours before harvesting the cells and isolating RNA for library preparation (Fig 6A). Differential expressed genes were screened with DESeq2 by comparing all against the control group (PBS). HK1 WT and HK1 LMP1 EVs statistically upregulated about 179 transcripts and downregulated 235 transcripts in the MCF10A cells comparing with the control (Fig 6B) (S3 Data). Heatmap representation showed individual genes were differentially expressed among the three different groups (Fig 6C). Comparison of the differentially expressed transcripts between cells treated with HK1 WT EVs and HK1 LMP1 EVs revealed 80 upregulated and 87 downregulated transcripts (S3A Fig). Canonical pathways analysis of the differentially expressed genes showed an enrichment and potential function in cell cycle, cellular growth and proliferation, cell death and survival and cancer (Fig 6D). Cellular expression of LMP1 has been shown to induce cell growth,

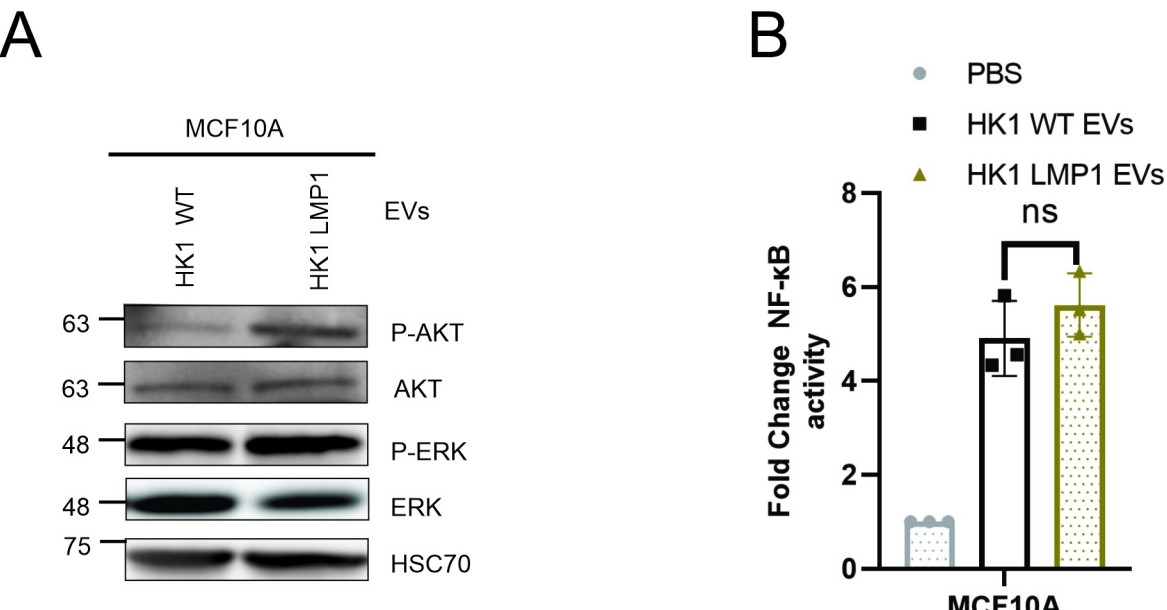

**Fig 5. LMP1 modified EVs activate ERK and AKT pathways.** (A) MCF10A cells were treated with equal protein of either HK1 WT or HK1 LMP1 EVs and western blot analysis was to assess AKT/ERK pathway activation. (B) MCF10A cells expressing the NF-kB luciferase reporter were exposed to PBS, HK1 WT or HK1 LMP1 EVs to evaluate activation of the NF-kB pathway.

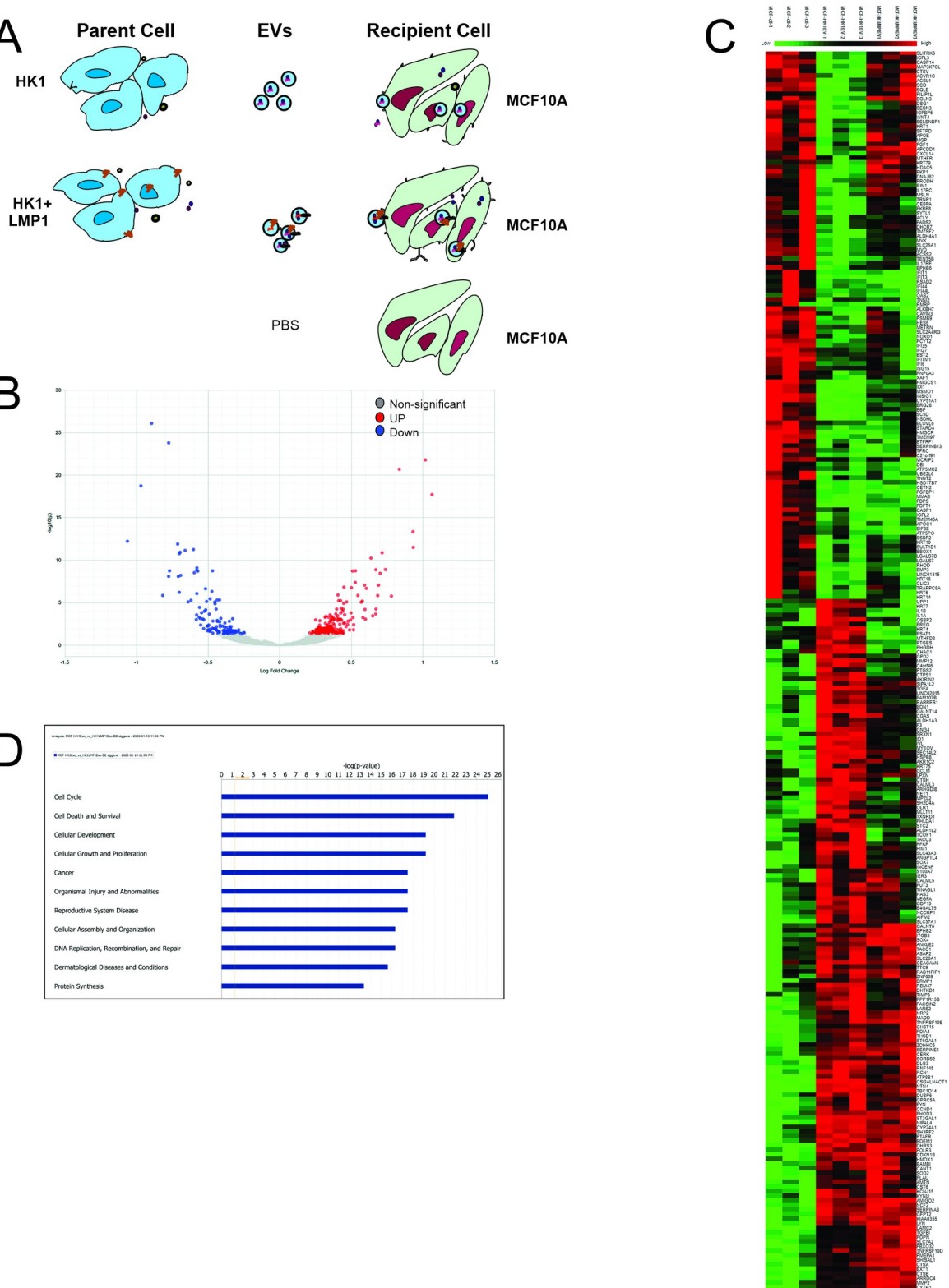

**Fig 6. HK1 WT and HK1 LMP1 EVs reprogram gene expression of recipient cells.** (A) Schematic model of protocol used to treat MCF10A cells with EVs. MCF10A cells were treated with equal number of EVs from either HK1 WT or HK1 LMP1 or equivalent volume of PBS every 24 hrs for 2 days before isolating mRNA for RNA-seq. (B) Volcano plot showing upregulated, downregulated and non-significant differentially expressed genes for the MCF10A cells exposed to HK1 WT or HK1 LMP1 EVs compared to cells exposed to PBS. (C) Heatmap representation of the differentially expressed genes from the cells treated with HK1 WT or HK1 LMP1 EVs compared to cells treated with PBS. (D) Canonical pathways using Ingenuity Pathway Analysis (IPA) of differentially expressed genes.

promote cell survival and suppress cell apoptosis depending on levels of expression [70]. Further analysis of the differentiated expressed genes clustered into four groups based on their expression level comparing to the control group (Table 1).

Furthermore, we performed global gene analysis expression comparing MCF10A cells exposed to HK1 WT or HK1 LMP1 EVs. Comparison of the two treatments revealed the following pathways to be downregulated: cell cycle, RNA transport, ribosome and proteasome (S4 Data). LMP1 containing EVs upregulated the following pathways in MCF10A cells: ECM-receptor interaction, focal adhesion, pathways in cancer and P13K-AKT signaling pathways (S4 Data). Most of these upregulated pathways play major role in ECM remodeling. In terms of EBV associated cancers this is important because it helps to start understanding how LMP1-modified EVs rewire the recipient cells gene expression towards premetastatic phenotype. Heatmap representation of the genes expressed in the ECM-receptor interaction pathways revealed noticeable individual gene expression between the treatment groups (Fig 7A). Additional analysis of the different individual components of ECM including cadherins, fibronectin, integrins and MMPs showed more than two-fold increase in gene expression comparing with HK1 WT and HK1 LMP1 EV treatment group (Fig 7B–7D). These identified genes enlighten the important roles LMP1 modified EVs might be playing in tumorigenesis and metastasis of EBV-associated cancers. Finally, using Ingenuity Pathway Analysis (IPA), we mapped a possible interactome model between those differentially expressed genes (S3B Fig). Taken together, the RNA-seq data highlight how LMP1 modified EVs reprogram the recipient cells by changing the expressed genes.

## LMP1 modified EVs upregulates MMPs and EMT associated gene expression

Fibronectin plays an important role in organization of the interstitial ECM and mediates cell attachments while the cadherins contribute to molecular characteristics of EMT [71–74]. MMPs are considered mainly to be responsible for the degradation of the ECM [75]. Cellular expression of LMP1 is proposed to affect the remodeling of the ECM by increasing expression

**Table 1. Analysis of the differentially expressed genes according to expression levels compared to control group.**

| Group | Untreated | HK1 EV | HK1-LMP1 EV | Enriched pathway |
|---|---|---|---|---|
| 1 | High | Low | High | Fatty acid metabolism, PPAR signaling pathway, Steroid biosynthesis. |
| 2 | High | Low | Low | Steroid biosynthesis, Metabolic pathways, Epstein-Barr virus infection. |
| 3 | Low | High | Low | AGE-RAGE signaling pathway, MAPK signaling pathway, TNF signaling pathway |
| 4 | Low | High | High | AGE-RAGE signaling pathway, ECM-receptor interaction, Focal adhesion, Epstein-Barr virus infection, PI3K-AKT signaling pathway |

Groups 2 and 4 represent genes which were overall regulated in the same way by both HK1 WT and HK1 LMP1 EVs. Groups 1 and 3 represent those genes which altered by the HK1 WT EVs but addition of the HK1 LMP1 EVs changed the expressed genes to a pattern like the control group. The differences noted in the different groups reflect the effects of LMP1 alone have on gene expression which might be through the different enriched cargo packaged in the EVs.

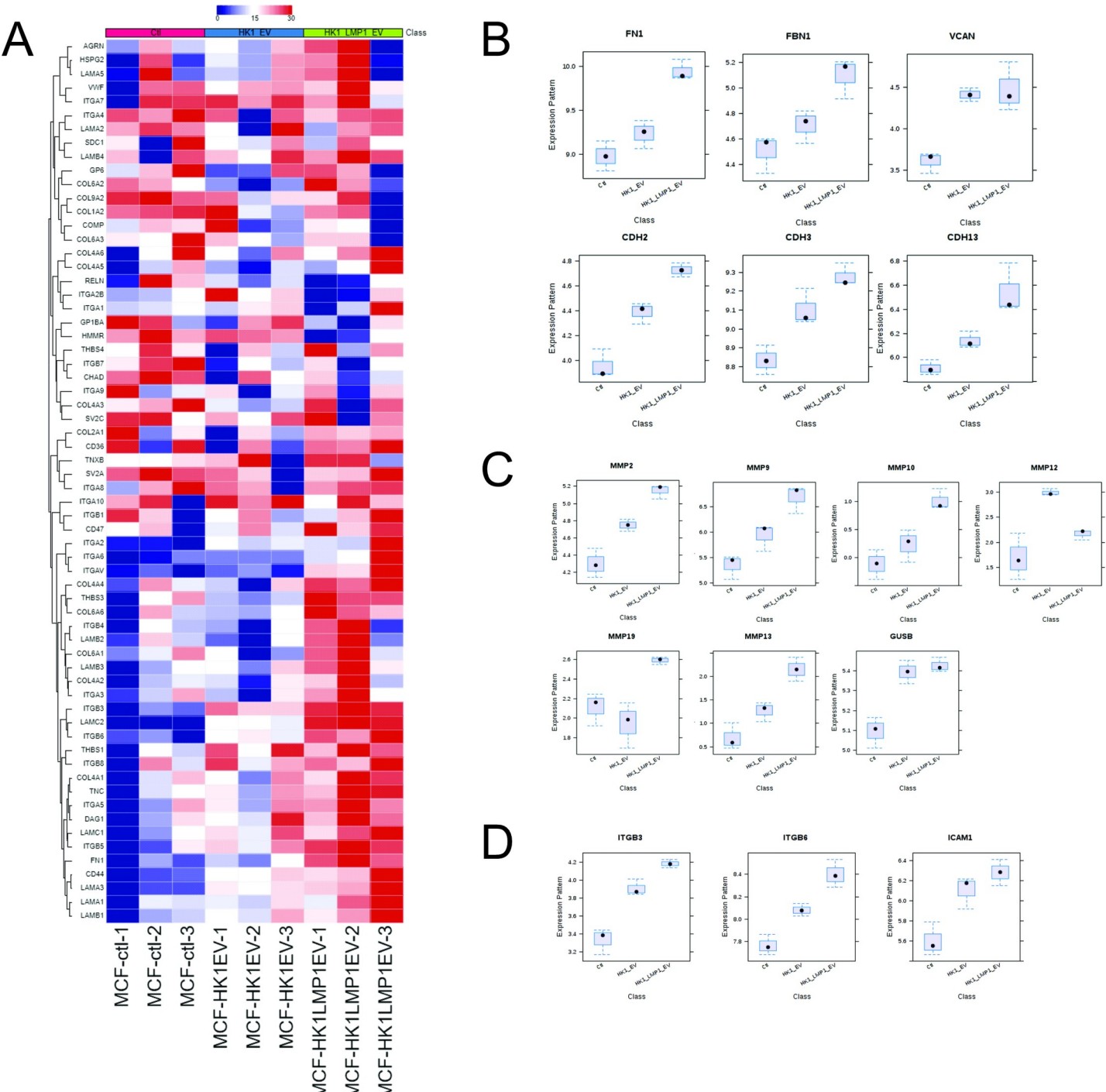

**Fig 7. LMP1 modified EVs rewire cells to promote a pro-metastatic phenotype.** (A) Heatmap representation of the global gene analysis of the ECM enriched pathway. (B-D) LMP1 modified EVs increase gene expression on cadherins, fibronectin and MMPs in the recipient cells.

of MMPs, fibronectin and inducing the EMT through the cadherin [73,76,77]. From our results we hypothesized that LMP1-modified EVs reprogram the recipient cells to promote the ECM remodeling. To assess whether LMP1 affects the protein expression of the different genes which might be involved in remodeling of the ECM, we performed immunoblot analysis

with equal protein loaded between cell and vesicle lysates from HK1 WT and HK1 LMP1 cells. The immunoblot analysis showed enrichment of fibronectin, integrinα5, N-cadherin, MMP9 and MMP2 in EVs compared to the whole cell lysates (Fig 8A and 8C). Furthermore, LMP1 containing EVs had an increased mRNA expression of the mentioned genes compared to the HK1 WT EVs. To verify the results obtained in (Fig 7B and 7C) where we observed that LMP1 containing EVs increased the expression of different genes in MCF10A cells, we performed RT-qPCR and immunoblot analysis on lysates collected from MCF10A cells after treating them with PBS, HK1 WT EV or HK1 LMP1 EVs. LMP1 increased mRNA expression of fibronectin, Nectin, alpha smooth muscle actin, claudin, integrinα5, N-cadherin, MMP9, and MMP2 and decreased expression of MMP12 (Fig 8B and 8D). LMP1 modified EVs also increased protein expression of fibronectin, integrinα5, MMP12 and MMP9 in MCF10A cells (Fig 8E). Treatment of HK1 cells with the EVs, revealed that the increase in gene expression by LMP1 was dose-dependent (S4A Fig). These results demonstrate the potential of LMP1 modified EVs and the packaged cargo to alter gene expression in the recipient cells. After uptake of the EVs by these naïve cells, the EVs can initiate transcription upregulation of different genes which could drive the cells towards a new phenotype. Collectively, upregulation of fibronectin, integrinα5, αSMA, N-cadherin and E-cadherin by LMP1 is associated with EMT and the MMPs expression is associated with degradation of ECM by the recipient cells and therefore aiding in the remodeling the tumor microenvironment [23,29,30,32,73,75].

## LMP1 modified EVs enhance cell attachment through integrinα5

Expression of LMP1 has been shown to induce EMT and its associated cell adhesion, motility and invasion features [20,26,74]. Elevated expression of fibronectin, an ECM protein enhances the cell adhesion and motility [21,71,78]. The major fibronectin receptors are α5β1 integrins which are expressed highly on surface of LMP1 containing cells [20,79]. Using the xCelligence system to monitor cell adhesion, it was recently shown that inhibition of integrinα5 by its neutralizing antibodies decreased LMP1-mediated enhancement of MCF10A cell adhesion [20]. Taken together, these results made us hypothesize that LMP1 containing EVs may be responsible for promoting the enhanced cell adhesion through integrinα5. To assess the effects of LMP1 modified EVs on ECM cell adhesion, the xCelligence system was used to monitor cell attachment of the MCF10A cells as described in literature [20,59]. The surfaces of the culture wells were either coated with fibronectin or uncoated and the seeded cells were either exposed to PBS, HK1 WT EVs or HK1 LMP1 EVs. Fibronectin increased attachment of MCF10A cells exposed to either PBS or HK1 WT EVs (Fig 9A and 9B). However, no changes in cell attachment were noted in the cells exposed to LMP1 modified EVs seeded in the fibronectin coated surfaces or the uncoated surfaces (Fig 9A and 9B). Conversely, our results showed that HK1 cells expressing LMP1 had an enhanced cell attachment at the end of 2 hours when exposed to fibronectin coated surfaces compared to the uncoated surfaces (S5A Fig). Surprisingly, comparative data analysis showed that in fibronectin coated surfaces, HK1 cells expressing LMP1 and MCF10A cells exposed to LMP1 modified EVs had similar levels of attachment at the end of 2 hours (S5B Fig). These findings support the idea that the LMP1 modified EVs contain or increase expression of the surface receptors which mediate the ECM cell adhesion. To determine this, the EVs were exposed to neutralizing antibodies to integrinα5 before incubating with the cells to evaluate cell attachment in fibronectin coated surfaces. The neutralizing antibodies to integrinα5 inhibited attachment of cells exposed to either the LMP1-modified EVs or HK1 WT EVs compared to the cells treated with the mouse IgG control antibody (Fig 9C and 9D). In HK1 cells expressing LMP1, the neutralizing antibodies to integrinα5 did not reduce cell attachment in comparison to cells treated with the control antibody (S5C and S5D

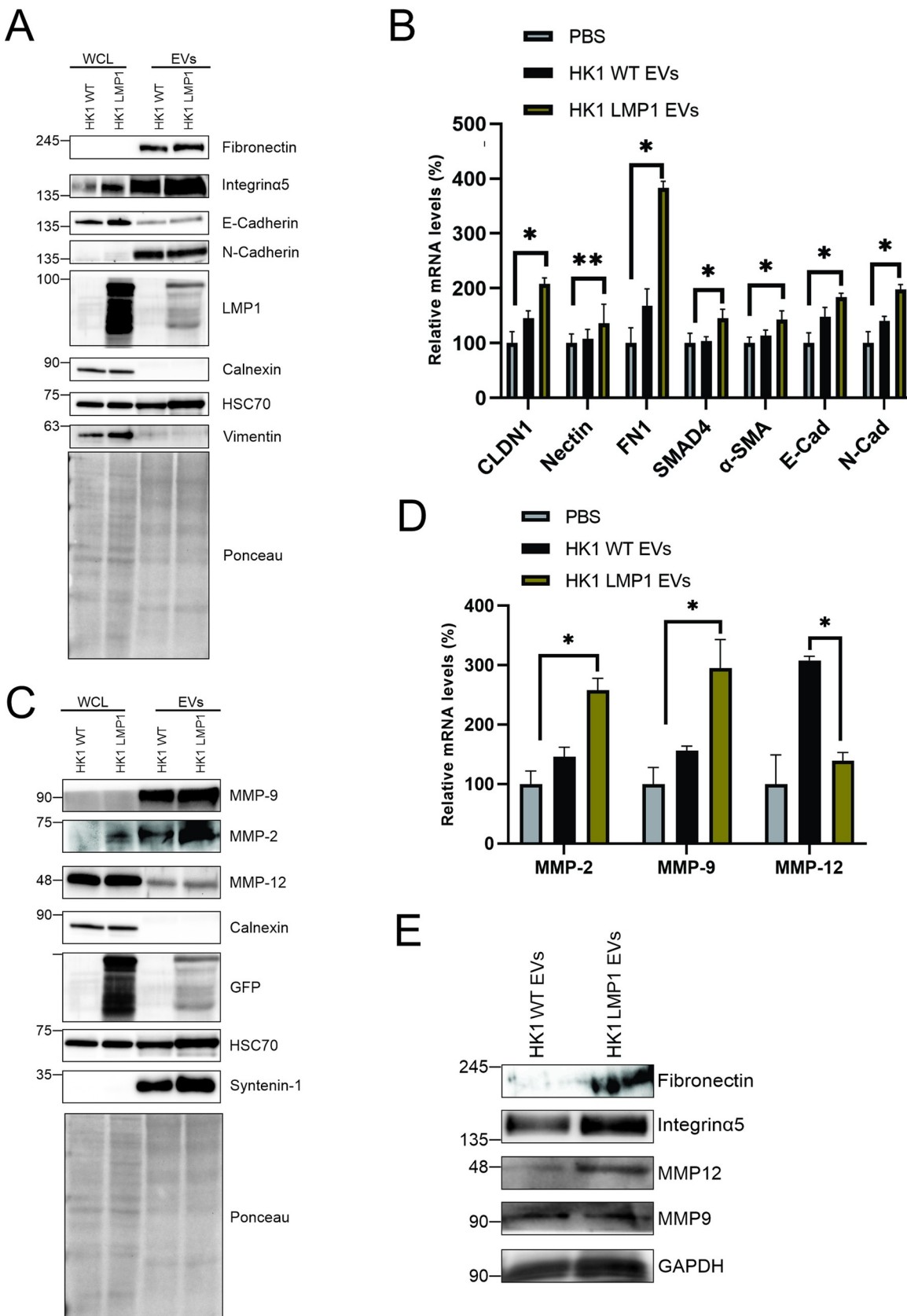

**Fig 8. LMP1 modified EVs upregulate MMPs and EMT associated gene expression.** (A, C) Immunoblot analysis of cell and vesicle lysates from HK1 or HK1 expressing LMP1 showing protein expression of cadherins, fibronectin, integrins and MMPs. (B, D) mRNA was collected from MCF10A cells treated with PBS, HK1 WT or HK1 LMP1 EVs and subjected to RT-qPCR to verify the observed RNA-seq data results. *, P < 0.05. (E) Immunoblot analysis of the cell lysates from MCF10A cell treated with either HK1 WT EVs or HK LMP1 EVs.

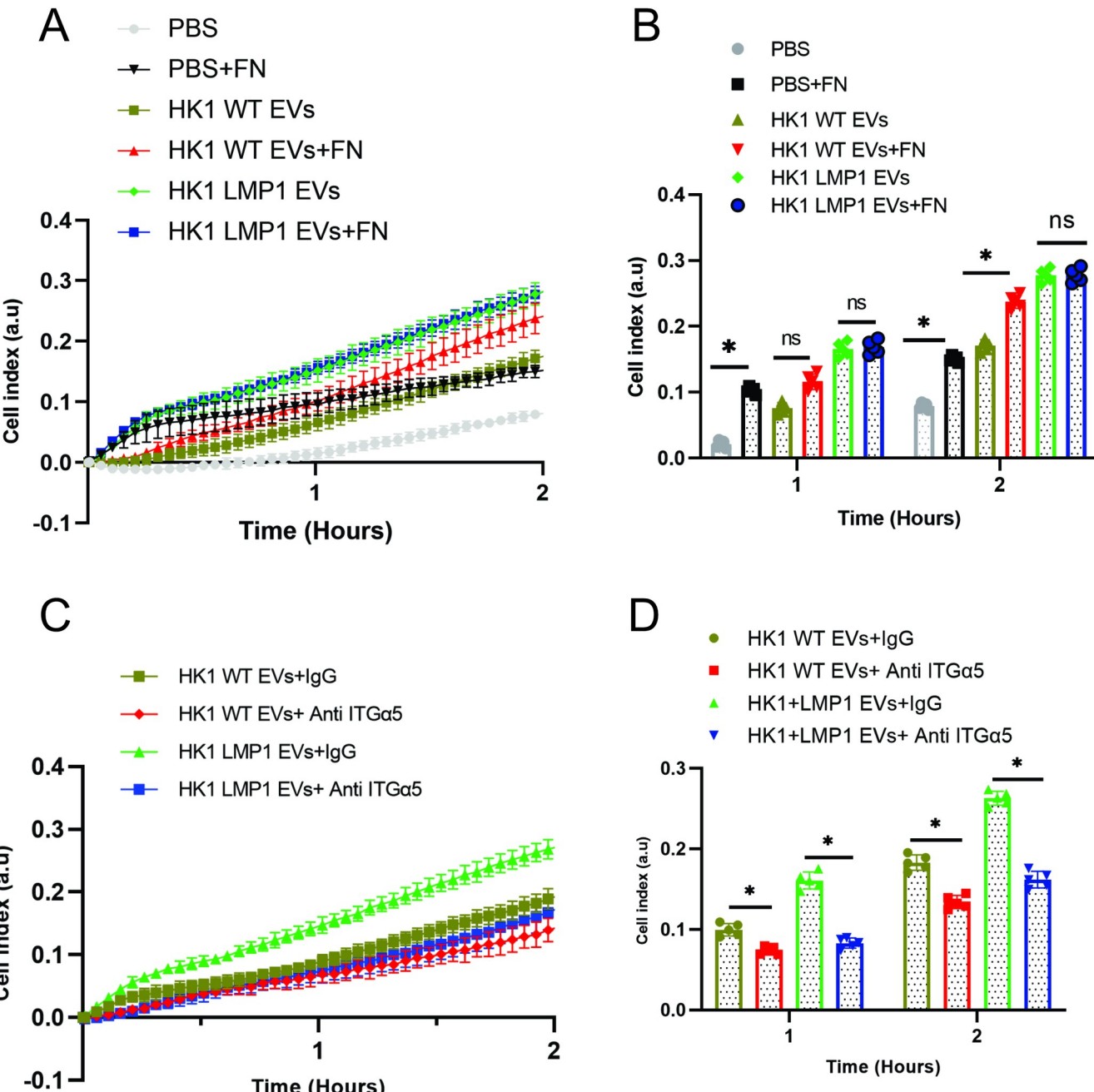

**Fig 9. LMP1 modified EVs enhance cell attachment through integrinα5.** MCF10A cells treated with PBS, HK1 WT or HK1 LMP1 EVs were evaluated for cell adhesion using impedance technology. (A-B) Comparison of the treated MCF10A cell attachment to fibronectin (FN) coated surfaces and uncoated surfaces. (C-D) Effect of integrinα5 neutralizing antibodies on the treated MCF10A cells on the fibronectin coated surfaces. Bar charts (B, D) show the indicated time points of corresponding growth curve (A, C). *, P < 0.05.

Fig). This is in contrast to what has been described for non-transformed MCF10A cells stably expressing LMP1 [20]. This could be due to the fact that the transformed HK1 cells high levels of surface exposed integrinα5 or other fibronectin binding proteins. Regardless, the neutralizing antibodies were able to block the enhanced attachment of MCF10A cells exposed to LMP1 modified EVs. Taken together, these findings begin uncovering the mechanisms in which LMP1 modified EVs mediate remodeling of the ECM and its receptor interactions which might affect other downstream processes.

## LMP1 modified EVs enhance MMP activity and promote cell invasion

Matrix metalloproteinases (MMPs) play an important role in tumor progression by degrading and remodeling the ECM. Expression of LMP1 has been shown to induce MMP1 and MMP9 implicating the viral oncoprotein in contributing to tumor metastasis [29,30]. Our data showed that LMP1 containing EVs differentially upregulated and increased mRNA expression of MMP2, MMP9 and downregulated expression of MMP12 (Fig 8D). These data made us speculate that LMP1 containing EVs enhances the MMP activity and functionality of the EVs. To evaluate that upregulation of MMPs by the LMP1 modified EVs is related with function, fluorometric MMP substrate was used to test pan-MMP activity in HK1 or HK1 LMP1 conditioned medium and isolated EVs. The results showed that conditioned media from HK1 LMP1 cells had significantly higher MMPs activity compared to the media from HK1 WT cells, even though they may contain similar level within the cytoplasm (Fig 10A). Furthermore, LMP1 modified EVs exhibited higher MMP activity compared to HK1 WT EVs (Fig 10B). To assess whether the enhanced MMP activity translates to increase in degradation of ECM, we performed cell invasion assay using the Xcelligence system. To mimic ECM, the floor of the upper chamber of the CIM plates were coated with a monolayer of Matrigel. HK1 cells expressing LMP1 showed enhanced invasion capacity compared to HK1 WT cells (S6A and S6B Fig). To determine if EVs promote the invasion potential of recipient cells, MCF10A cells were exposed to PBS, HK1 WT EVs or HK1 LMP1 EVs. The results showed that cells exposed to PBS were not able to degrade the matrix (Fig 10C and 10D). However, LMP1-modified EVs increased the invasion potential of the cells in comparison to cells exposed to the HK1 WT EVs (Fig 10C and 10D). Taken together, these findings begin to show underlying mechanisms through which LMP1 modified EVs can remodel the tumor microenvironment by degrading the ECM and hence increasing the metastatic capabilities of the EBV associated cancers.

## Discussion

Numerous studies have now established the major role EVs play in pathogenesis of different diseases and their huge potential as biomarkers and therapeutic targets. In case of viral-modified EVs, they have been shown to promote development and advancement of different cancers through modulation of the tumor microenvironment [57,59,80,81]. Transfer of LMP1 containing EVs to naïve recipient cells is proposed to mediate transfer of functional pro-metastatic vesicles to induce cancer development and promote progression to surrounding and distant cells. This study begins to uncover mechanisms underlying how LMP1 modified EVs mediate tumor microenvironment remodeling to enhance metastatic properties. LMP1-modified EVs reprogram the recipient cell gene expression towards a metastatic phenotype driving enhanced cell attachment, migration, and invasion.

Previously we demonstrated that EBV and KSHV latently infected cells release EVs with distinct content and this correlates to levels of LMP1 expression in EV producing cells [51]. Mass spectrometry data analysis showed that differentially expressed EV proteins isolated from EBV and KSHV infected primary effusion lymphoma cells expressing LMP1 had unique

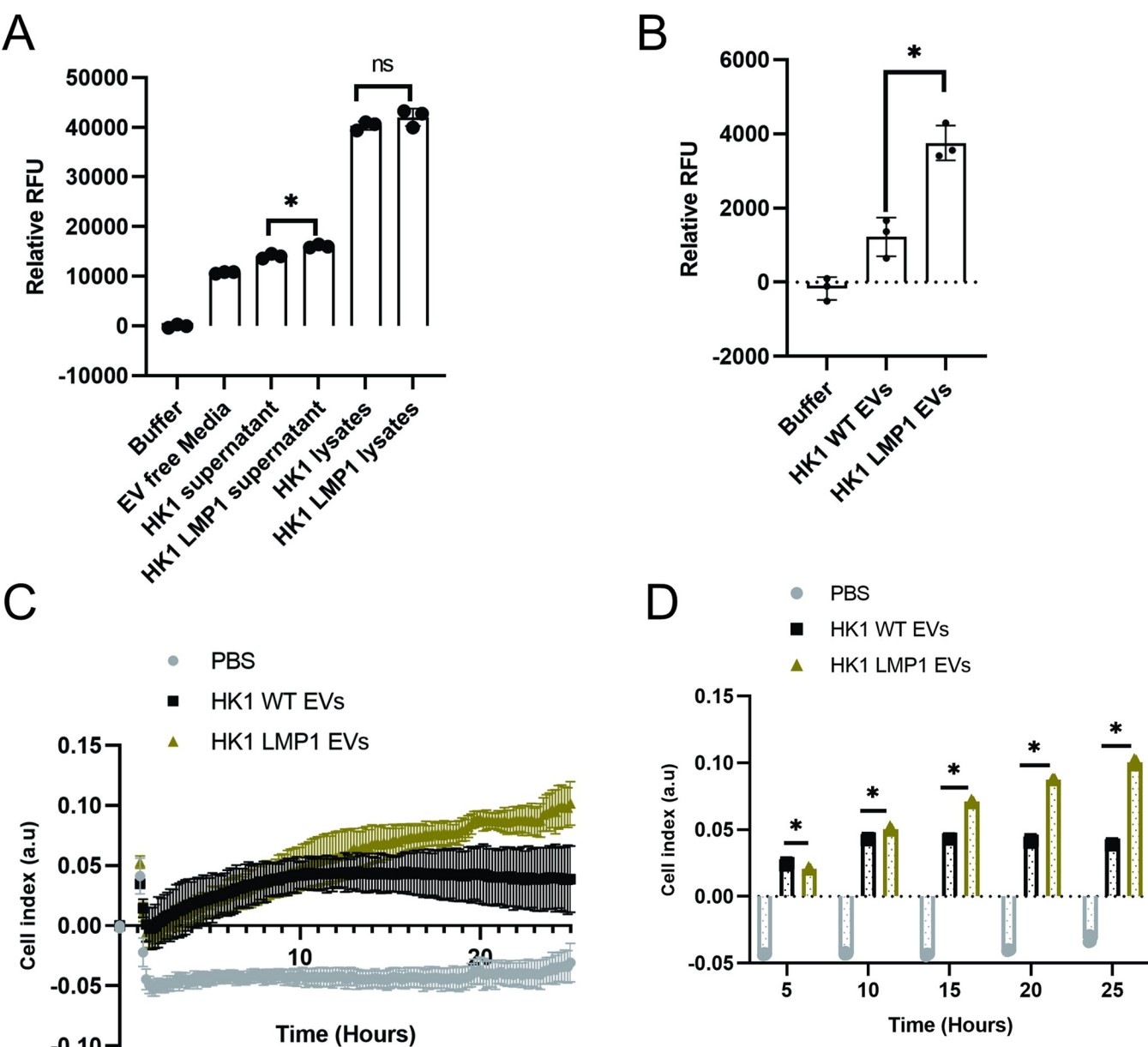

**Fig 10. LMP1 modified EVs enhance MMP activity and promote cell invasion.** (A-B) Fluorogenic MMP substrate was utilized to measure the pan-MMP activity of HK1 WT or HK1 LMP1 conditioned media and isolated EVs. (C-D) Cell invasion through the Matrigel of the EV treated MCF10A cells was assayed using the xCelligence system. *, P < 0.05.

clustering pattern compared to those EVs isolated from cells not expressing LMP1. This work highlighted that viral proteins may have a major impact on the EV proteome. In case of LMP1 modified EVs, these data alluded that the EVs released might be unique and with distinct biological properties affecting the cross talk between near or distant cells. Our current results confirm that LMP1 in the absence of other viral proteins is alone able to dramatically alter the protein cargo and functions of the EVs. Interestingly, LMP1 also modulated the microRNA and mRNAs packed in the EVs. Different mechanisms have been suggested to drive protein trafficking to EVs. Previous studies have shown plasma membrane targeting, higher order oligomerization and protein modification like myristoylation, prenylation, and palmitoylation

are responsible for EV targeting [82–84]. LMP1 harbors palmitoylation, prenylation and myristoylation motifs that can facilitate efficient packaging of different proteins. Additionally, LMP1 has been shown to localize and signal from lipid rafts which are membrane domains associated with proteins and molecules to facilitate their release in EVs [85,86]. Mutations within LMP1 transmembrane domains has been suggested to mediate negative and positive regulatory elements for EV sorting of LMP1 [82]. The engagement of LMP1 in multiple pathways involved in EV cargo sorting and release suggests that MP1 might be facilitating incorporation of these proteins. Mechanisms of whether there is an existence of preferential targeting of certain proteins or cargoes are still not well understood. Furthermore, LMP1 activated signal transductions can also enhance mRNA and protein expression of different EV biogenesis and secretion genes which might lead to the modification of the EV proteome.

Different studies have demonstrated the functional properties of the LMP1 modified EVs. LMP1 containing EVs promote cell attachment, proliferation, migration, invasion potential, and radio resistance of cells [32,56,57]. Distinct cargo packaged into the EVs means transfer of the LMP1 modified EVs can transmit different proteins, lipids and miRNAs to recipient cells which produces diverse phenotypes. More importantly, these LMP1 containing EVs mediate the transfer of signaling molecules to recipient cells where they induce NF-kB, PI3K/AKT and MAPK/ERK signaling pathways resulting in downstream effects like increased proliferation, migration and invasion. LMP1 has been shown to activate all these pathways but mechanisms responsible for the downstream effects are poorly understood. Abrogation of LMP1 trafficking to EVs by knocking out CD63 has been shown to increase LMP1-induced noncanonical NF-κB and ERK activation intracellularly [49,52]. Recently, LMP1 has been shown to enhance EV release through Syndecan-2 and synaptotagmin-like-4 through NF-κB signaling and this promotes cell proliferation, invasion and tumor growth *in vivo* [54]. KSHV EVs have also been shown to activate ERK1/2 which leads endothelial cell migration and proliferation [59]. Pathway analysis of the LMP1 upregulated proteins in EVs showed enrichment in the signaling pathways including MAPK signaling pathway, TNF, NF-kappa B and HIF-1 signaling. These data support the concept that LMP1 modified EVs are signaling competent when transported to naïve recipient cells leading to the activation of LMP1-specific pathways that likely contribute to the cell transformation process. Furthermore, these LMP1 containing EVs possess other potent signaling factors like EGFR, AKT, FGF-2, Ezrin and HIF1α which also affect cell proliferation, migration and invasion of the recipient cells [24,32,48].

Our results demonstrate that LMP1 modified EVs reprogram the recipient cells gene expression towards a pre-metastatic phenotype. This study was designed to mimic short exposure response of the recipient cells to the LMP1 containing EVs. Prior studies have shown that transfer of Kaposi's Sarcoma-associated herpesvirus EVs to recipient cells induce transcriptome rewiring leading to cell proliferation and migration [59,80]. Yogev et al. also demonstrated that KSHV viral encoded miRNAs released in EVs function in uninfected recipient cells to cause metabolic reprograming and this might be used by other tumor viruses [80]. EVs derived from EBV infected cells have also been shown to modulate the metabolism of the recipient cells in a similar fashion to KSHV EVs [51,80]. Data presented here demonstrate that LMP1 alone can promote metabolic rewiring through EVs transfer to recipient cells. LMP1 alters the protein and miRNAs packaged into EVs and these likely function to induce the metabolic reprogramming. The LMP1 packaged EVs alter the metabolic status of the recipient cells by activating aerobic glycolysis and autophagy leading to reverse Warburg effect [57,87]. LMP1 mediated signaling affects glycolytic flux by enhancing plasma membrane translocation of glucose transporter 1 (GLUT1), increasing the first glycolysis pathway enzyme, hexokinase 2 and upregulates monocarboxylate transporter 4 (MCT4) [57,88,89]. Furthermore, LMP1-mediated aerobic glycolysis in NPC tumor microenvironment has been associated with

immune escape [87]. In the case of LMP1, the altered EVs might be regulating metabolism for evasion of the immune system and hence promote remodeling of the tumor microenvironment. Collectively, the transcriptional reprograming by the LMP1 containing EVs may similarly be inducing a reverse Walberg effect which leads to downstream effects like enhanced cell proliferation, migration and invasion. One limitation of our study is that we cannot rule out the possibility that cytokines or other soluble factors produced by recipient cells in response to LMP1 modified EVs may contribute to the transcriptional reprogramming of the cells.

Global gene pathway analysis comparing transcripts from MCF10A cells treated with HK1 WT EVs or LMP1 modified EVs showed significant transcriptional reprogramming changes in ECM-receptor interactions and focal adhesions. Cellular expression of LMP1 has been shown to induce EMT and upregulate MMP9 to degrade the ECM which leads to ECM remodeling [23,30,31,73,78]. Our results support the hypothesis that ECM remodeling is EV-mediated. Aga et al. previously showed that LMP1 increases levels of HIF1α in EVs and the LMP1 modified EVs induce the cadherin switch associated with EMT [32]. HIF1α has also been shown to be an alternative way to induce expression of fibronectin and cell migration through TGF-β1 [26]. LMP1 upregulates fibronectin expression and increases cell surface expression of its receptors α5β1 to accelerate invasion and metastasis [26]. Our data show that LMP1 containing EVs mediate enhanced cell attachment and invasion through integrinα5 and MMPs, respectively. LMP1 modified EVs express surface receptors integrinα5 and MMPs which can be mediate the enhanced cell attachment and invasion if they bind to the cell surface of recipient cells. Alternatively, the LMP1 modified EVs are taken up by the recipient cells and initiate transcriptional upregulation of the different genes leading to protein expression. EV adsorption has been shown to plateau within hours; however, after 15–30 minutes of incubation of EVs and cells, EVs can be detected attached to cell surface and some are found in the lumen or associated with phagocytic membranes [90]. The crosstalk and cellular response between the recipient cells once they have been exposed to these LMP1 containing EVs warrants further investigation. Taken together, our results begin to show the significant role of LMP1 modified EVs play in mediating ECM remodeling to facilitate cell adhesion, motility and invasion.

In conclusion, results presented here provide new insights into the mechanism of how the EBV oncoprotein LMP1 mediates the transcriptional rewiring of the recipient cells towards a new phenotype due to the altered EV cargo and content. Furthermore, these results show possible biomarkers or therapeutic targets which can be helpful in treatment of EBV-associated cancers.

## Methods

### Cell and media

HK1 (a gift from George Tsao, Hong Kong University) and HK1 cells expressing inducible GFP-LMP1, pQCXP GFP-LMP1 CTAR1 and pQCXP LMP1 CTAR2 have been previously described [49,91]. To induce LMP1 expression in the HK1 GFP-LMP1 cells, we added doxycycline to a final concentration of 1 μg/μL. The cell lines were grown in RPMI-1640 cell culture medium (Lonza; 12-702Q) with corresponding supplements added. The cells were maintained at 37˚C with 5% $CO_2$. The media was supplemented with a 10% final concentration of fetal bovine serum (FBS; Seradigm; 1400–500), 2 mM L-glutamine (Corning; 25-005-Cl), 100 IU penicillin-streptomycin (Corning; 30-002-CI), and 100 μg/mL:0.25 μg/mL antibiotic/antimycotic (Corning; 30-004-CI). MCF10A cells were maintained in Dulbecco's modified Eagle's medium along with equivalent supplements as specified by ATCC (Lonza; MEGM kit; CC-3150) and Cholera toxin to a final concentration of 100 ng/mL.

## Extracellular vesicle isolation and enrichment

EVs were isolated from approximately 500 mL of cell culture media. The medium was centrifuged at $500 \times g$ for 10 minutes in an Eppendorf 5804R using an S-4-104 rotor to pellet out the cells. The supernatant was then passed through a 0.22 μm VWR vacuum filtration (76010–402). The conditioned media was further concentrated by Minimate TFF Capsules (100KDa cut-off, Pall, OA100C12) to 50 mL. The remaining 50 mL supernatant was split in 50mL conical tubes where 1:1 volume of 16% (2X) polyethylene glycol (average $M_n$, 6000; Alfa Aesar; 25322-68-3) and 1 M sodium chloride was added to samples and incubated overnight at 4˚C. The incubated samples were centrifuged at $3{,}214 \times g$ for 1 hour in an S-4-104 rotor. The pellet from each tube was then washed with 1mL of 1X phosphate-buffered saline (PBS), centrifuged at $100{,}000 \times g$ for 70 min in a Beckman MAX-E using the TLA120.2 rotor. The collected EV Samples were resuspended in about 300 μL of particle-free PBS for nanoparticle tracking analysis or protein quantification to be used in functional assays and RNA-seq. For mass spectrometry analysis, HK1 WT EVs and HK1 LMP1 EVs were harvested using a modified ExtraPEG method followed by purification using iodixanol density gradient as previously described [60,91,92]. We have submitted all relevant methodological details of our experiments to the EV-TRACK knowledgebase (EV-TRACK ID: EV200071) [93].

## Nanoparticle tracking analysis (NTA)

Nanoparticle tracking was performed using a Malvern NanoSight LM10 instrument, and videos were processed using NTA 3.4 software as previously described [49,92].For NTA processed data, see S5 Data.

## Immunoblot analysis

Equal protein of EVs was subsequently resuspended in 2× Laemmli sample buffer (4% SDS, 100 mM Tris, pH 6.8, 0.4 mg/mL bromophenol blue, 0.2 M dithiothreitol [DTT], 20% glycerol, 2% β-mercaptoethanol [BME]). Whole cell were processed as previous described and the lysate were mixed with 5× Laemmli sample buffer (10% SDS, 250 mM Tris, pH 6.8, 1 mg/mL bromophenol blue, 0.5 M DTT, 50% glycerol, 5% BME) to a final concentration of 1X and boiled at 95˚C for 10 mins. An equal amount of protein was loaded into an 10% SDS-PAGE gel for electrophoresis and then transferred to a nitrocellulose membrane. The blots were blocked in Tris-buffered saline solution containing 0.1% Tween-20 and 5% non-fat dry milk. The primary antibodies used included Alix (Santa Cruz; Q-19), HSC70 (Santa Cruz; B-6), TSG101 (Santa Cruz; C-2), CD81 (Santa Cruz; SC-9158), syntenin-1 (Santa Cruz; SC-100336), Fibronectin (Santa Cruz; SC-73611), E-cadherin (Cell Signaling; 3195S), N-Cadherin (Santa Cruz; SC-7939), Vimentin (Santa Cruz; SC-6260), integrinα5 (Cell Signaling; 4705S), MMP2 (Cell Signaling; 87809), MMP12 (Santa Cruz; SC-390863), MMP9 (Cell Signaling; 13667), GFP (Rockland; 600-101-215), Flotillin-2 (Santa Cruz; H-90), CD63 (Abcam; TS63), Calnexin (Santa Cruz; H-70), GAPDH (Genetex; 100118), integrin β3 (Cell Signaling; 13166S), integrinαV (Cell Signaling; 4705S), integrinβ1 (Cell Signaling, 9699), EGFR (Santa Cruz; SC-03).The blots were probed with horseradish peroxidase (HRP)-conjugated secondary antibodies: rabbit anti-mouse IgG (Genetex; 26728), rabbit anti-goat IgG (Genetex; 26741) or goat anti-rabbit IgG (Fab fragment) (Genetex; 27171). The blots were incubated with Pico ECL (Thermo; 34080). The blots were imaged using an Image Quant LAS4000 (GE life science) and processed with ImageQuant TL v8.1.0.0 software, Adobe Photoshop CS6, and CorelDraw Graphic Suite 2019.

## NF-kB luciferase cell reporter assay

The pHAGE lenti-NF-κB-luc-GFP has been previously described [94]. The lentiviral particles were generated by transfecting HEK293T cells together with packaging plasmids as previously described [65]. The lentiviral particles were used to transduce MCF10A cells to generate a stable cell line. The subsequent stable cells were selected using medium containing puromycin (2 μg/mL) for 2 weeks.

MCF10A cells expressing the NF-kB luciferase cell reporter were seeded into a 24 well plate at $10^5$ per well. 24 hours post seeding the cells were treated with HK1 WT EV, HK1 LMP1 EV at 10 μg/well or equivalent volume of PBS and media was changed to serum free. Cell lysates were harvested after 24 hours of treatment with the EVs. Dual luciferase reporter assay system (Promega, E1910) was used. Passive cell lysis was done according the manufacturers protocol. This was followed by reading the assay on luminometer per protocol.

## Mass spectrometry

Equal protein of EV lysate was run and separated on a 4–20% SDS PAGE (Lonza; 59111) as previously described [65,95]. After the separation, the gel fixed and stained with the Coomassie stain before being fractionated. The excised gel pieces containing the proteins were reduced, then alkylated before trypsin digestion [95–97]. The eluted peptides from the gel pieces were submitted to FSU Translational Science Laboratory to be analyzed on the Thermo Q Exactive HF (High-resolution electrospray tandem mass spectrometer) as previously described [65]. Resulting raw files were searched with Proteome Discoverer 2.4 using SequestHT, Mascot and Amanda as search engines. Data files are available via ProteomeXchange with identifier PXD021914. Scaffold (version 4.10) was used to validate the protein and peptide identity. Peptide identity was accepted if Scaffold Local FDR algorithm demonstrated a probability greater that 95.0%. Likewise, protein identity was accepted if the probability level was greater than 99.0% and contained a minimum of two recognized peptides as previously described.

## Bioinformatic enrichment analysis

The proteins identified to be increased 2-fold or higher by LMP1 were subjected to the bioinformatic analysis. Cellular compartment enrichment and biological processes was done through FunRich v3.1.3 [98]. The Kyoto Encyclopedia of Genes and Genomes (KEGG) pathway analyses were conducted using the NetworkAnalyst 3.0 [99]. Upregulated and downregulated pathways were analysed on iDEP [100].

## Cell adhesion, proliferation, migration and invasion assays

Cell proliferation, adhesion, migration and invasion were measured using the xCelligence RTCA DP instrument (ACEA Biosciences, San Diego, CA, USA). For cell proliferation and adhesion MCF10A cells were dissociated using trypsin and the cells (10,000) were seeded in the 16 well plates (E-16 plate, 5469830001 ACEA Biosciences). The MCF10A cells were incubated together with HK1 WT or HK1 LMP1 EVs to a final concentration of 10μg/ml when seeding in the wells to monitor proliferation and adhesion using electrical impedance. The chambers were pre-coated with fibronectin final concentration at 10 μg/mL. For the cell adhesion blocking experiments, the EVs were initially preincubated with 20 μg/mL of Integrinα5 neutralizing antibody for about 30 minutes at 4˚C or equivalent amounts of the Mouse IgG control. Impedance measurements when done every 3 minutes for about 3 hours for the adhesion assay and 50 hours for the cell proliferation assay. A minimum of four wells for each sample were measured. For cell migration assays, MCF10A cells were also prepared as before and

incubated with either HK1 WT EVs or HK1 LMP1 EVs before being seeded into a CIM-16 plate. The lower chamber was filled with media containing 1–2% FBS to act as a chemoattractant. The cells (40,000) were seeded in the upper chamber in serum free media and readings were taken every 10 mins for 24 hours. For the invasion assays, Matrigel was diluted with precooled serum free media to a concentration of 800 μg/mL and about 50 μL was initially added to each upper well chamber of the CIM-plate 16. Next, we removed about 30 μL of the Matrigel solution from each well leaving 20 μL in each well to coat the membrane surface. The upper chamber containing Matrigel was placed in tissue culture incubator (37 °C) for about 4 hours for polymerization. MCF10A cells (40,000) exposed to either HK1 WT EVs, PBS or HK1 LMP1 EVs were added to the wells and impedance was measured every 15 minutes for 24 hours. The cell index shows the degree of cell adhesion, proliferation and migration.

## Next-generation sequencing

HK1 WT and HK1-LMP1 EV associated miRs were isolated and sequenced as stipulated in the protocol below. Each sample was performed in triplicate. EV samples were treated with RNase (ThermoFisher; AM2294) to final concentration of 50 ng/mL, at room temperature for 30 mins. RNase inhibitor (NEB; M0314) and PCR grade water were added to EV sample to total volume of 200 μL. miRs were isolated by adding 600 μL Trizol LS (ThermoFisher; 10296010) according to manufacturer's instruction. To increase the yield of small RNAs, two volume of 100% ethanol and linear acrylamide (VWR; 97063–560) were used instead of isopropyl alcohol and incubation time was also increased to overnight at -20˚C. The isolated RNAs were quantified by Qubit microRNA assay kit (ThermoFisher; Q32880). Small RNA libraries were generated with NEBNext Multiplex Small RNA Library Prep Set for Illumina (NEB; E7300). To increase yield and prevent primer/adaptor dimer, 3' SR primer was diluted to 1:2 and increase ligation time to overnight at 16˚C. The library was ran on Bioanalyzer with HS DNA chip (Agilent; 5067–4626) for quality control and quantified by KAPA library qPCR kit (KAPA; KR0405). Then the libraries were pooled at equal molar amounts and submitted to the Florida State University College of Medicine Translational laboratory for sequencing on illumina NovaSeq 6000 system.

For RNA-seq experiments, MCF10A cells were grown in 12 well plates with 1 ml of media. 24 hours post seeding, media was changed to serum free and physiological concentrations (10E+12) [101] of HK1 WT EVs, HK1 LMP1 EVs or equivalent volume of PBS was added. New EVs were added every 24 hours for 48 hours prior to the cell harvest. RNA-seq for EV treated MCF10A cell were performed with the following kits. RNAs from triplicate cell samples were isolated by Trizol and incubated with DNase I to remove trace genomic DNA contamination. Then, mRNA library was prepared by combination of NEBNext Ultra RNA Library Prep Kit (NEB, E7530) and NEBNext Poly(A) mRNA Magnetic Isolation Module (NEB, E7490). Similar to miR library, HS DNA chip and KAPA library kit were used before submitting to sequencing by illumina NovaSeq 6000.

## RNA-seq data analysis

Raw data for miR-seq were submitted to OASIS [102] online miR analysis tool to identify small RNAs on Human reference genome hg38. Differential expressed miRs from HK1-LMP1 EVs were analyzed by both OASIS and miRNet using default settings.

RNA-seq data was analyzed by NetworkAnalyst 3.0. Genes with counts less than 10, variance less than 10% and unannotated were filtered and normalized by Log2-counts per million. Differential expressed gene were identified by DEseq2. Heatmap of globe differential expressed genes and gene enriched pathways were also visualized by the same online tool. IPA (Qiagen)

software was used to generate potential interaction map between DE genes in the ECM pathway.

Several different tools, including IPA, OmicsNet and MIENTURNET, were used to predict miR-mRNA interaction network in EV treated MCF10A cells. The data discussed in this publication have been deposited in NCBI's Gene Expression Omnibus [103] and are accessible through GEO Series accession number GSE155202. (https://www.ncbi.nlm.nih.gov/geo/query/acc.cgi?acc=GSE155202).

## RNA isolation and reverse transcription

Total RNA of cell or EV samples were isolated by Trizol reagent (ThermoFisher; 15596018) and quantified by nanodrop. Less than 1 μg of total RNA was used for reverse transcription by qScript cDNA SuperMix (Quantabio; 95048). cDNAs were store in -20˚C until further use.

## Quantitative real-time PCR (RT-qPCR) and data analysis

Standard 3-step cycles protocol (40 cycles of 95˚C for 5 s, 60˚C for 10 s, 72˚C for 20 s) was used in all qPCR reactions. PerfeCTa SYBR Green FastMix (Quantabio; 95072–012), assay primers and cDNA of cell or EV were prepared in 20 μL reaction and run on CFX96 qPCR machine (Bio-Rad). Gene expression level were first normalized to the housekeeping gene GAPDH and then calculated with ΔΔCt method. See Table 2 below for primers sequence used for qPCR.

## MMP activity assay

MMP activity in cell lysate, conditioned medium or EVs were tested by fluorogenic pan-MMP substrate (R&D; ES001). The cell for test was lysis by RIPA buffer without proteinase inhibitor

**Table 2. qPCR primer sequence.**

| Gene ID (Aliases) | Sequence | Reference or Primer Bank ID [104,105] |
|---|---|---|
| FN1 (Fibronectin) | ACTGAGACTCCGAGTCAGCC | PMID: 26479443 |
| | TTCCAACGGCCTACAGAATT | |
| CLDN1 (Claudin-1) | CCTCCTGGGAGTGATAGCAAT | 296785063c1 |
| | GGCAACTAAAATAGCCAGACCT | |
| MMP-2 | GATACCCCTTTGACGGTAAGGA | 189217851c3 |
| | CCTTCTCCCAAGGTCCATAGC | |
| MMP-9 | GGGACGCAGACATCGTCATC | 74272286c3 |
| | TCGTCATCGTCGAAATGGGC | |
| MMP-12 | CATGAACCGTGAGGATGTTGA | 261878521c1 |
| | GCATGGGCTAGGATTCCACC | |
| Nectin-1 | CTGCAAAGCTGATGCTAACC | DOI: 10.1128/JVI.01582-16 |
| | GATGGGTCCCTTGAAGAAGA | |
| CDH2 (N-Cadherin) | AAATTGAGCCTGAAGCCAAC | PMID: 26479443 |
| | GTGGCCACTGTGCTTACTGA | |
| CDH1 (E-Cadherin) | AAAGGCCCATTTCCTAAAAACCT | 169790842c3 |
| | TGCGTTCTCTATCCAGAGGCT | |
| SMAD4 | CTCATGTGATCTATGCCCGTC | 195963400c1 |
| | AGGTGATACAACTCGTTCGTAGT | |
| ACTA2 (α-SMA) | CTATGAGGGCTATGCCTTGCC | 213688378c3 |
| | GCTCAGCAGTAGTAACGAAGGA | |
| GAPDH | GGAGCGAGATCCCTCCAAAAT | 378404907c1 |
| | GGCTGTTGTCATACTTCTCATGG | |

and analyzed immediately. EV and cell lysate sample were first quantified by OD660, and 10 μg total protein was used in MMP assay for all samples. Conditioned and control cell culture medium were pre-cleared by centrifuge at $500 \times g$ for 10 mins and 50 μL volume used in reaction. All the sample was diluted with reaction buffer (100mM Tris pH7.5, 100mM NaCl, 10mMCaCl$_2$, 0.05% NP40) to 100 μL volume containing 10 μM substrate. The assay plates were incubated at room temperature for indicated time and then read on fluorescent reader with 320nm excitation/405nm emission.

### Transmission electron microscopy

Isolated EVs were prepared for electron microscopy imaging as previously described [49].

## Supporting information

**S1 Table. Predicted protein targets for the microRNA enriched pathways.**
(XLSX)

**S1 Fig. Bioinformatic analyses of the LMP1 downregulated proteins.** (A) Pathways (KEGG) and (B) biological processes analysis. (C) immunoblot analysis verification of proteins identified in mass spectrometry.
(TIF)

**S2 Fig. LMP1 expressing HK1 cells enhance cell adhesion.** (A) HK1 WT cells, HK1 cells expressing LMP1 cells or HK1 WT cells exposed to equal protein of HK1 WT, HK1 LMP1 EVs were seeded into xCelligence E-16 plate. Cell attachment was monitored for about 2 hours. (B) HK1 cells expressing LMP1 C-terminal activating region 1 promote cell attachment. (C) Comparative analysis of cell attachment between HK1 cells expressing LMP1 and MCF10A cells treated with EVs. (D) LMP1 modified EVs promotes cell attachment in a dose dependent manner. *, $P < 0.05$.
(TIF)

**S3 Fig. Predicted genes involved in the LMP1 interactome.** (A) Volcano plot showing upregulated, downregulated and non-significant differentially expressed genes for the MCF10A cells exposed to HK1 LMP1 EVs compared to HK1 WT EVs. (B) Model of possible predicted interaction of the genes LMP1 modified EVs promote expression.
(TIF)

**S4 Fig. mRNA gene expression in HK1 cells is dose dependent.** mRNA was collected from HK1 cells treated with different amounts (25μg vs 50μg) of PBS, HK1 WT or HK1 LMP1 EVs and subjected to RT-qPCR to verify the observed RNA-seq data results. *, $P < 0.05$.
(TIF)

**S5 Fig. LMP1 expressing cells cell adhesion is not affected integrinα5 neutralizing antibodies.** HK1 cells expressing LMP1 were evaluated for cell adhesion using impedance technology. (A) Comparison of the LMP1 expressing cell attachment to fibronectin (FN) coated surfaces and uncoated surfaces. (B) Comparative analysis between LMP1 expressing cells and MCF10A cells treated with EVs. (C-D) Effect of integrinα5 neutralizing antibodies on LMP1 expressing cells on the fibronectin coated surfaces. *, $P < 0.05$. (A-B) Cell invasion capacity through the Matrigel of the HK1cells expressing LMP1 was assayed using the xCelligence system. (C) Comparative analysis of cell invasion potential between LMP1 expressing cells and MCF10A cells treated with EVs. *, $P < 0.05$.
(TIF)

**S6 Fig. LMP1 expressing cells promote cell invasion.** (A-B) Cell invasion capacity through the Matrigel of the HK1cells expressing LMP1 was assayed using the xCelligence system. (C) Comparative analysis of cell invasion potential between LMP1 expressing cells and MCF10A cells treated with EVs. *, P < 0.05.
(TIF)

**S1 Data. Proteomic analyses of HK1 WT EVs and HK1 LMP1 EVs.** All proteins identified during mass spectrometry comparing HK1 WT EVs and HK1 LMP1 EVs.
(XLSX)

**S2 Data. MicroRNAs differentially expressed between HK1 WT EVs and HK1 LMP1 EVs.**
(XLSX)

**S3 Data. Raw RNA-seq data from MCF10A cells treated with HK1 WT EVs or HK1 LMP1 EVs.**
(XLSX)

**S4 Data. KEGG Pathway analysis of the genes identified during the RNA-seq.**
(XLSX)

**S5 Data. NTA data processing of the EVs harvested.**
(XLSX)

## Acknowledgments

We thank Monica Abou Harb for her help in preparing samples and transmission electron microscope grids for imaging, FSU National High magnetic Field Laboratory staff for the help in acquiring the TEM images. FSU Translational Science Laboratory for help in mass spectrometry data acquisition, and RNA seq.

## Author Contributions

**Conceptualization:** Dingani Nkosi, David G. Meckes, Jr.

**Data curation:** Dingani Nkosi, Li Sun, Leanne C. Duke, David G. Meckes, Jr.

**Formal analysis:** Dingani Nkosi, Li Sun, Leanne C. Duke, David G. Meckes, Jr.

**Funding acquisition:** David G. Meckes, Jr.

**Investigation:** Dingani Nkosi, Li Sun, Leanne C. Duke, David G. Meckes, Jr.

**Methodology:** Dingani Nkosi, Li Sun, Leanne C. Duke, David G. Meckes, Jr.

**Project administration:** David G. Meckes, Jr.

**Resources:** David G. Meckes, Jr.

**Software:** Dingani Nkosi, Li Sun, David G. Meckes, Jr.

**Supervision:** David G. Meckes, Jr.

**Validation:** David G. Meckes, Jr.

**Visualization:** Dingani Nkosi, Li Sun, David G. Meckes, Jr.

**Writing – original draft:** Dingani Nkosi, Li Sun, David G. Meckes, Jr.

**Writing – review & editing:** Dingani Nkosi, Li Sun, Leanne C. Duke, David G. Meckes, Jr.

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
