## [Decision Letter · Decision Letter 0]

25 Jun 2020

Dear Dr Meckes,

Thank you very much for submitting your manuscript "Epstein-Barr virus LMP1 manipulates the content and functions of extracellular vesicles to enhance metastatic potential of recipient cells" for consideration at PLOS Pathogens. As with all papers reviewed by the journal, your manuscript was reviewed by members of the editorial board and by three independent reviewers. All three reviewers found the manuscript to be interesting and important, and the experiments to be well designed and executed. Despite this enthusiasm, the reviewers pointed to several experiments that will require additional benchwork, hence the editorial decision reached. In light of the reviews (below this email), we would like to invite the resubmission of a significantly-revised version that takes into account the reviewers' comments. Key revisions include validation by Western blot of some of the key proteins determined by Mass spec, including a method to correlate EV and RNA quantitation, and providing a positive control for the MMP activity data. Reviewer 1 also found the data on ERK activation to be unconvincing, while Reviewer 3 requested that more effort be made to perform some dose dependent experiments. Reviewer 2 included several minor editorial recommendations, and asked that details on accessions and acquisitions be added. The suggestion by Reviewer 1 that work with LMP1 mutants should be included is sound, and this would indeed further strengthen the manuscript, but I consider that to be an optional, not essential, addition to a revised manuscript.

We cannot make any decision about publication until we have seen the revised manuscript and your response to the reviewers' comments. Your revised manuscript is also likely to be sent to reviewers for further evaluation.

Sincerely,

Ashlee V. Moses

Associate Editor

PLOS Pathogens

Erik Flemington

Section Editor

PLOS Pathogens

Kasturi Haldar

Editor-in-Chief

PLOS Pathogens

orcid.org/0000-0001-5065-158X

Michael Malim

Editor-in-Chief

PLOS Pathogens

orcid.org/0000-0002-7699-2064

Reviewer's Responses to Questions

**Part I - Summary**

Reviewer #1: The current study by Nkosi, Sun and Meckes titled “Epstein-Barr virus LMP1 manipulates the content and functions of extracellular vesicles to enhance metastatic potential of recipient cells” describes the effect of LMP1 from EBV in Extracellular Vesicles (EVs). LMP1 containing EVs promoted cell growth, migration, differentiation, and regulate immune cell function. The authors hypothesize that LMP1 alter EV content and the recipient cells by activation of cell signaling pathways and increased gene expression. Here, they show that LMP1 expression alters the EV protein and microRNA content packaged into EVs, and these EVs enhance recipient cell adhesion, proliferation, migration, invasion as evident by the activation of ERK, AKT, and NF-κB signaling pathways. The EV purification was a combination of tangential flow filtration (TFF) followed by precipitation using PEG-6000 and ultracentrifugation. They also further characterized these EVs using markers including CD63, CD81, Alix, TSG101, Syntenin-1, HSC70 and Calnexin. They find that LMP1 can modify the EV proteome (MAPK1, PTEN, GSK3B and CRKL regulation) and alter the miRNAs packaged into the EVs and they show a pro-migratory phenotype in recipient cells. Overall, I found the manuscript interesting, focused and carefully designed.

Reviewer #2: This manuscript on EBV LMP1 and EVs suggests that EV content changes with infection and can promote metastasis. This a nice body of work with well-designed experiments. Writing is good but could be more concise. I have a few comments that might help to improve the piece.

Reviewer #3: This is an extension of a previous work done by Dr. Meckes lab suggesting that LMP1-modified EVs reprogram recipient cells towards a pre-metastatic phenotype.

Several piece of data support that the expression profile of cells receiving EVs from LMP1-expressing cells changes and genes related to cell survival, cell proliferation, and various metabolic pathways are affected.

Although several solid piece of data have been presented a few control experiments suggested below will strengthen this work.

**Part II – Major Issues: Key Experiments Required for Acceptance**

Reviewer #1: There are however few concerns including:

1. Some of the significant pieces of data from the proteomics (Figure 2) need to be validate with WBs. This is to show full-length proteins and potential validation of some key proteins from Mass Spec.

2. Data in Figure 3 is interesting, but does not correlate with the number of EVs to the number of the RNAs. The authors should consider using few miRNAs (or other RNAs) that could serve as control markers for copy number and then back calculate the number of the isolated RNAs (piRNA, miRNA, etc) to get a sense of their significance in downstream functional assays.

3. Some of the data in Figure 5A (WB) does not look convincing and needs to be redone.

4. It is surprising not to use any mutants of LMP1 for specificity of function, especially for data in Figures 9 and 10.

Reviewer #2: Please deposit raw RNAseq data with appropriate databases and provide accessions.

Please include NTA acquisition and analysis details in the manuscript.

Reviewer #3: -Experiments in which LMP1 appears to modify MMP activity to promote cell invasion and cell attachment through integrins lack a positive control. This control is important to evaluate the significance of these changes that look moderate. In addition, comparisons with MCF10A cells that express LMP1 will allow for a better evaluation.

- Some dose dependent experiments are important to be included.

- It is unclear how many EVs have been used in most assays and for how long.

-It is unclear what is the percentage of EVs carrying LMP1.

-Is there a change at the protein level of fibronectin in LMP1-expressing cells and in cells exposed to EVs carrying LMP1?

**Part III – Minor Issues: Editorial and Data Presentation Modifications**

Reviewer #1: None

Reviewer #2: For graphs showing data for which statistical comparisons are done: would prefer dots rather than bar graphs as better indication of variability.

Recommended for replication purposes: deposit EV methods with EV-TRACK (evtrack.org) and give the accession citing van Deun, et al, Nat Methods, 2017.

"Secrete" in my understanding means release from within the cell, e.g. through endosomal pathways. Unless that is the intent, would recommend "release" as more general.

Further editing for accuracy and concision is recommended. For example, "Pivotal to these purification methods is the ability to harvest EVs that retain all the biological properties EVs supposedly encompass," could be better rendered as "An ideal purification method would retain EV function(s)."

Reviewer #3: -There is a mislabeling of the panels in Fig. 6

PLOS authors have the option to publish the peer review history of their article (what does this mean?). If published, this will include your full peer review and any attached files.

Reviewer #1: No

Reviewer #2: No

Reviewer #3: No
---

## [Editor Report · Decision Letter 1]

2 Oct 2020

Dear Dr Meckes,

We are pleased to inform you that your manuscript 'Epstein-Barr virus LMP1 manipulates the content and functions of extracellular vesicles to enhance metastatic potential of recipient cells' has been provisionally accepted for publication in PLOS Pathogens.

Best regards,

Ashlee V. Moses

Associate Editor

PLOS Pathogens

Erik Flemington

Section Editor

PLOS Pathogens

Kasturi Haldar

Editor-in-Chief

PLOS Pathogens

orcid.org/0000-0001-5065-158X

Michael Malim

Editor-in-Chief

PLOS Pathogens

orcid.org/0000-0002-7699-2064
---

## [Editor Report · Acceptance letter]

9 Dec 2020

Dear Dr. Meckes,

We are delighted to inform you that your manuscript, "Epstein-Barr virus LMP1 manipulates the content and functions of extracellular vesicles to enhance metastatic potential of recipient cells," has been formally accepted for publication in PLOS Pathogens.

Best regards,

Kasturi Haldar

Editor-in-Chief

PLOS Pathogens

orcid.org/0000-0001-5065-158X

Michael Malim

Editor-in-Chief

PLOS Pathogens

orcid.org/0000-0002-7699-2064